# Learning to Orchestrate Agents in Natural Language with the Conductor

**Stefan Nielsen** [1*]  **Edoardo Cetin** [1*]  **Peter Schwendeman** [2*†]  **Qi Sun** [1 3]
**Jinglue Xu** [1]  **Yujin Tang** [1]

[1] Sakana AI, Japan    [2] University of Michigan, USA    [3] Institute of Science Tokyo, Japan

## Abstract

Powerful large language models (LLMs) from different providers have been expensively trained and finetuned to specialize across varying domains. In this work, we introduce a new kind of *Conductor* model trained with reinforcement learning to automatically discover powerful coordination strategies among LLMs. Our Conductor learns not only to design targeted *communication topologies* for effective agent-to-agent collaboration, but also to *prompt engineer* focused instructions to the LLMs to maximally leverage their individual capabilities. We show that, by learning optimal coordination strategies over pools of powerful worker LLMs, a 7B Conductor achieves significant performance gains beyond any individual worker, attaining state-of-the-art results in challenging reasoning benchmarks, such as LiveCodeBench and GPQA. By training with randomized agent pools, our conductor effectively adapts to arbitrary sets of open- and closed-source agents, meeting any user requirements. Furthermore, allowing the Conductor to select itself as a worker gives rise to recursive topologies, elevating performance with a new form of dynamic test-time scaling through online iterative adaptation. More broadly, ours is among the early work demonstrating language model coordination can be unlocked through RL, where powerful coordination strategies emerge naturally in LLMs through pure end-to-end reward maximization.

## 1 Introduction

Through unprecedented scale and engineering effort, modern Large Language Models (LLMs) (Anthropic, 2025; OpenAI, 2025; 2023; Team et al., 2023) demonstrate the ability to solve formidably complex tasks, with performance even approaching that of top human experts (Luong & Lockhart, 2025). These remarkable latent capabilities are essentially the product of training scale and better utilization of the models themselves, with a history of research showing the importance of combining and developing the two for effective utilization (Liu et al., 2023; Brown et al., 2020). However, utilizing these latent capabilities to their full potential remains a challenge even for experienced users, with manually-designed agentic workflows being critical components of commercial AI products (AWS, 2025; Anysphere, 2025; Microsoft, 2025) while effective prompting and

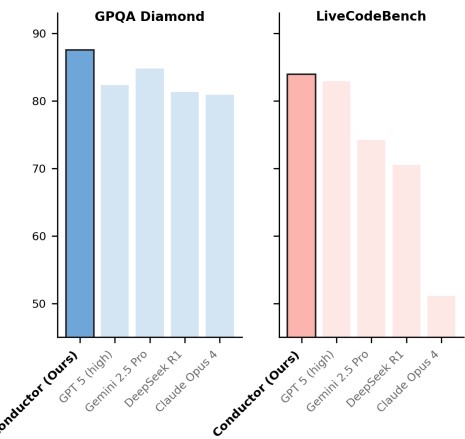

Figure 1: Our Conductor attains the state-of-the-art in GPQA and LiveCodeBench.

self-refinement strategies are still a core focus of the current research (Wei et al., 2022; Madaan et al., 2023a). Furthermore, different models are finetuned to specialize in particular datasets and domains – with no single LM universally optimal across all tasks (Chang et al., 2024).

---

*Equal contribution
†Work done during internship at Sakana AI
Correspondence to: {stefannielsen,edo,yujintang}@sakana.ai

Based on these considerations, we introduce *the RL Conductor*: a new kind of reasoning model trained with reinforcement learning (RL) (Guo et al., 2025; Shao et al., 2024) to dynamically divide challenging problems, delegate targeted subtasks, and design communication topologies for a set of worker LLM agents. Our model is itself an LLM tasked to output a sequence of *workflow steps*, each defined by a natural language instruction focusing on some aspect of the overall task, the assigned agent receiving that instruction, and each agent's visibility to the other agents as they perform their role. This formulation enables our Conductor to construct entirely flexible agentic workflows customized to each input problem, with common strategies such as prompt engineering, refinement, and even meta-prompt optimization, naturally emerging from end-to-end reward maximization.

By effectively leveraging the complementary skills of its powerful worker agents, a 7B Conductor attains state-of-the-art results on challenging reasoning benchmarks such as LiveCodeBench and GPQA Diamond (Figure 1). Our systematic evaluation shows that the performance of our Conductor goes considerably beyond both traditional self-reflection strategies with any of its workers, and also prior costly multi-agent collaboration baselines that use a much larger number of agent calls (Wang et al., 2024; Yue et al., 2025). These findings hold well beyond the set of training tasks and across a wide range of math, coding, and natural science domains, demonstrating the potential of our model to supersede manual agentic scaffolds with a general and robust end-to-end approach.

We also effectively extend our framework by finetuning our pretrained Conductor with two additional techniques to better suit custom user requirements and further push performance in exchange for test-time compute. First, by training with randomized agent pools at each step, we show our model can generalize to arbitrary sets of open and closed-source workers, allowing users to still harness state-of-the-art performance with no expensive API calls. Furthermore, by allowing the Conductor to specify itself as a worker LLM, we give rise to a new kind of *recursive topology*, which unlocks a new tunable axis of inference-time scaling in reasoning models.

To facilitate future research and extensions, we share the codebase of our new framework with this submission, allowing the reproduction of all results. In summary, our contributions are threefold:

- We introduce the RL Conductor, a language model trained through end-to-end reinforcement learning to divide challenging problems, delegate targeted subtasks, and design communication topologies for a set of worker LLMs – all in natural language.

- We demonstrate that by obtaining effective prompt engineering and coordination skills, a small 7B Conductor can raise its worker LLMs to new heights, attaining state-of-the-art results on complex reasoning tasks and outperforming more expensive multi-agent baselines.

- We show how a short finetuning unlocks extensions such as adaptability to arbitrary agent pools and powerful recursive topologies that yield a new test-time scaling axis.

## 2 REINFORCEMENT LEARNING AND REASONING

Recent progress in scaling the performance of LLMs by increasing test-time compute has been driven by the reinforcement learning (RL) "reasoning" paradigm, which has established itself as a ubiquitous new stage of training large-scale open and closed source models (Jaech et al., 2024; Guo et al., 2025; Meta AI, 2025; Yang et al., 2025; Comanici et al., 2025). The high-level recipe, introduced by the DeepSeek R1 line of work (Wang et al., 2023; Shao et al., 2024; Guo et al., 2025), optimizes an LLM $\pi_\theta$, using a custom system prompt, by making it generate its own completions $o_i$ to a set of verifiable problems $D = (q_1, s_1), \ldots, (q_N, s_N)$. This custom system prompt instructs the model to answer each question while providing a thinking trace before its solution attempt, placing each inside appropriate `<think>` and `<solution>` tags. The rewards $r_i$ for each output of the model are then determined by two conditions:

1. The format condition, setting $r_i$ to -1 for any of the model's outputs that do not adhere to the specified `<think>`/`<solution>` format.

2. The correctness condition, setting $r_i$ to 1 in case the model's correctly formatted outputs match the solution $s_i$ and to $-0.5$ otherwise.

Using these rewards, the model is trained with GRPO (Shao et al., 2024), a simple online RL algorithm. GRPO uses the LLM $\pi_\theta$ to generate a set of $G > 1$ grouped completions $\{o_1, \ldots, o_G\}$

**User**

> *We call a subarray of an array complete if the number of distinct elements in the subarray is equal to the number of distinct elements in the whole array. Return the number of complete subarrays.*

**Conductor**

> *Here's the plan: 1) Model 2 will develop an algorithm to efficiently count all possible complete subarrays for a given array, 2) Model 0 will implement the function to solve the problem.*
>
> *model_id = [ 2, 0 ]*
> *subtasks = ["Develop an efficient algorithm to count the number of complete subarrays of an array", "Implement the algorithm described by the previous agent in Python"]*
> *access_list = [ [] , ["all"] ]*

Figure 2: **The Conductor output.** The Conductor responds with the entire coordination strategy.

for each question $q \in D$. Then, for $\beta \geq 0$ and a KL-divergence penalty to the reference model $\mathbb{D}_{\mathrm{KL}}(\cdot \| \pi_{\mathrm{ref}})$, the optimization objective is given by the KL-discounted policy maximization:

$$J(\theta) = \mathbb{E}_{q \sim D, \, \{o\}_1^G \sim \pi_\theta(\cdot|q)} \left[ \frac{1}{G} \sum_{i=1}^{G} \Big( \min\big(r_i A_i, \, \mathrm{clip}(r_i, \, 1-\epsilon, \, 1+\epsilon) \, A_i\big) - \beta \, \mathbb{D}_{\mathrm{KL}}(\pi_\theta \, \| \, \pi_{\mathrm{ref}}) \Big) \right],$$
(1)

using the grouped completions to compute a Monte-Carlo *advantage function* (Sutton et al., 1999):

$$A_i = \frac{r_i - \mathrm{mean}(\{r_1, \ldots, r_G\})}{\mathrm{std}(\{r_1, \ldots, r_G\})}.$$
(2)

As specified in its system prompt, this simple recipe has been shown to be effective at aligning the model with self-emergent thinking capabilities, yielding unprecedented task specialization.

## 3    LEARNING TO CONDUCT AN ORCHESTRA OF MODELS

In this work, we design a new reinforcement learning framework for training a *Conductor* language model to prompt-engineer and coordinate a set of much larger and more powerful LLM agents. The Conductor outputs full *agentic workflows* that divide an input task, allocate natural-language subtasks, and define targeted communication strategies to best make use of the agents' complementary capabilities – as detailed in the remainder of the section.

### 3.1    FRAMING AGENT COORDINATION IN NATURAL LANGUAGE

**The Conductor task.** The Conductor's objective is to solve tasks *indirectly* by designing different *agentic workflows* specific to any input question $q_i$.

> **Definition.** Each agentic workflow is defined as a sequence of *workflow steps* whose final output is returned as the actual Conductor response $o_i$. Each step specifies a string with a natural-language *subtask*, an integer id corresponding to the *assigned worker agent* responsible for performing that subtask, and an *access list* indexing which subtask solutions from the previous steps to include in the worker's context.

The information about each agentic workflow is parsed from the Conductor's response after its *chain-of-thought* as three simple Python lists with the same number of entries. This output structure is exemplified in Figure 2, in which the Conductor devises an agentic workflow first querying agent 2 to devise an algorithm, and then agent 0 to implement it in Python with the previous answer from agent 2 in context. To accelerate learning and make our framework compatible with arbitrary models, we provide the Conductor with detailed instructions in the system prompt alongside examples with the expected output format. This design lets the Conductor freely craft tailored subtasks and communication strategies across its workers, allowing the specification of agentic workflows

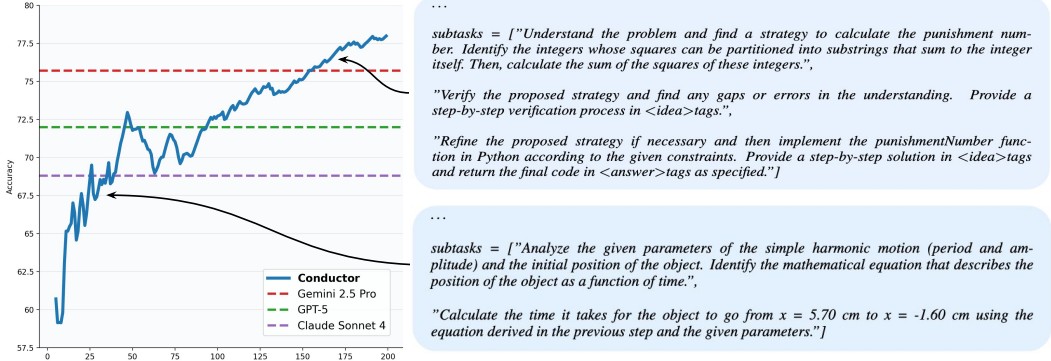

Figure 3: **Emergence of powerful coordination strategies over training.** Early in training, the Conductor issues sound subtasks, but does not tap useful collaborative strategies such as verification (bottom-right). Near convergence, the Conductor has learned to utilize planners, issue targeted instructions, instruct workers to share reasoning, and leverage verification and refinement (top-right), leading to the Conductor surpassing the worker agents' performance on our training dataset (left).

ranging from simple best-of-N and sequential chain-like topologies to parallelizable arbitrary tree-structured approaches, harnessing the individual strengths and synergies of its highly-specialized agents.

**Workflow execution and learning dynamics.** Each agentic workflow outputted by the Conductor is executed sequentially by prompting the specified worker agents with their assigned natural language subtask. In each workflow step, the worker's context includes the sequence of previous subtasks and corresponding responses defined in the access list, simply provided as past messages in a conversational template. Analogously to the traditional RL framework, the reward $r_i$ for each response from the Conductor model is determined by two progressive conditions:

1. The Conductor format condition, setting $r_i$ to 0 for responses from which the Python lists of subtasks, worker ids, and access lists cannot be parsed.

2. The Conductor correctness condition, setting $r_i$ to 1 if the final output from executing a well-formatted agentic workflow $o_i$ matches the solution $s_i$ and to $0.5$ otherwise.

While training end-to-end with the conductor reward is inherently compatible with any RL algorithm (Schulman et al., 2017; Ahmadian et al., 2024), in this work, we employ the GRPO formulation described in Section 2. Training the Conductor with this simple recipe, we observe the emergence of problem breakdowns and prompt-engineered subtasks that match the strengths of each worker, together with communication strategies that combine independent attempts with final debate rounds. As shown in Figure 3 and detailed in the following section, these Conductor behaviors lead to our model quickly surpassing each of its much larger workers, yielding state-of-the-art performance far beyond manually-designed multi-agent pipelines.

## 3.2 EXTENDING THE RL CONDUCTOR

**Adaptive worker selection.** To make the Conductor robust to variation in the available worker pool, we extend our framework to operate over customizable subsets of models. To this end, we simply finetune a pretrained Conductor, restricting it for each question to a randomly sampled $k$-model subset from the larger total pool of $n$ workers and accordingly modifying its input instructions. After training, this design makes the Conductor generalize and extract strong performance over a specific desired subset of $k \leq n$ models, allowing our model to cater towards specific user constraints and cost preferences. This extension aims to drive flexible coordination, with the Conductor learning to reconfigure problem breakdowns based on the varying synergies of arbitrary sets of agents.

**Recursive topologies and test-time scaling.** We also extend our framework by introducing *recursive* agentic workflows, allowing the Conductor to leverage its own capabilities to complement the other worker agents. During each inner *recursive call*, the Conductor is provided as an additional input its own *parent* output from which the call was instantiated, together with the previous agent's response. With this context, the Conductor will be given the chance to instantiate a new agentic

Table 1: **Comparison with previous best "unconstrained" results.** The Conductor's performance is significantly beyond official reported results across several challenging reasoning benchmarks, setting new records and pushing the boundary of LLM capabilities with collective intelligence.

| Model | In-Domain Tasks | | | | Unseen Tasks | | | |
|---|---|---|---|---|---|---|---|---|
| | M500 | MMLU | RLPR | LCB | AIME25 | BCB | GPQA-D | Avg. |
| gemma-3-27b-it | 39.8 | 81.3 | 16.67 | 13.14 | 20.7 | 14.86 | 38.4 | 32.12 |
| Qwen3-32B | 73.5 | 83.5 | 31.00 | 21.21 | 20.0 | 30.41 | 64.1 | 53.81 |
| Qwen3-32B (thinking) | 80.7 | 84.1 | 37.25 | 25.86 | 72.9 | 28.38 | 66.8 | 56.57 |
| R1-Distill-Qwen-32B | 82.5 | 84.4 | 33.50 | 26.86 | 63.0 | 33.07 | 58.1 | 54.49 |
| Claude Sonnet 4 | 96.0 | 91.4 | 36.70 | 46.54 | 74.3 | 37.16 | 77.7 | 65.69 |
| Gemini 2.5 Pro | 96.0 | 92.4 | 40.55 | 67.24 | 78.3 | 37.51 | 84.8 | 70.97 |
| GPT 5 | 99.0 | 93.5 | 42.20 | 82.90 | 90.8 | 32.75 | 82.3 | 74.78 |
| *Conductor* (**Ours**) | **99.4** | **94.1** | **44.75** | **83.93** | **93.3** | **37.86** | **87.5** | **77.27** |

workflow or end the coordination loop by returning the final subtask solution directly to the user. We avoid infinite recursion loops by allowing recursive calls, after the initial root Conductor call, to occur only up to a specified maximum number before returning the final response to the user. We unlock recursive capabilities in pretrained Conductor models by simply finetuning with the same RL algorithm while manually instantiating a single recursion call for half the samples in each batch. After training, this formulation allows for adaptively increasing the maximum number of recursion calls during inference to effectively introduce a new form of test-time scaling, leveraging recursion as a tunable compute axis beyond open-ended chain-of-thoughts. We present a visualization of recursion in Fig. 12.

## 4 EVALUATING THE RL CONDUCTOR

We evaluate the capabilities of the RL Conductor at scale. In this section, we quantitatively show how a 7B Conductor attains state-of-the-art results across current frontier models, outperforming a wide range of expensive self-reflection strategies and traditional multi-agent collaboration baselines at a fraction of the cost. Furthermore, we demonstrate how extensions such as test-time recursive scaling and adaptive worker selection can be efficiently integrated by finetuning our pre-trained Conductor to unlock new, powerful capabilities. Finally, we thoroughly analyze the properties and behavior of pretrained Conductors, illustrating the emergence of prompt engineering capabilities with increasing model size and difficulty-adaptive coordination strategies.

### 4.1 TRAINING AND EVALUATION SETUP

For our main experiments, we train a small Conductor with 7B parameters with the framework detailed in Section 3, starting from a Qwen2.5 checkpoint (Hui et al., 2024). Our Conductor is tasked to devise agentic workflows of up to five steps using both proprietary frontier models, such as Gemini-2.5-Pro (Comanici et al., 2025), Claude-Sonnet-4 (Anthropic, 2025), and GPT-5 (OpenAI, 2025), together with open-source alternatives such as DeepSeek-R1-Distill-Qwen-32B (Guo et al., 2025), Gemma3-27B-instruct (Team et al., 2025), and Qwen3-32B (Yang et al., 2025). Our training dataset comprises 960 problems from four reasoning domains covering a range of math, coding, and general real-world reasoning questions, selected for their difficulty and diversity. In composing this dataset, we used the seminal competition MATH corpus (Hendrycks et al., 2021), the multitask language comprehension task MMLU (Hendrycks et al., 2020), the real-world reasoning task RLPR (Yu et al., 2025), and the code generation benchmark LiveCodeBench V1 (Jain et al., 2024). We empirically find that, by relying on a powerful set of workers, our framework effectively sidesteps the canonical exploration problem faced by other small models trained with RL (Cetin et al., 2025), efficiently reaching convergence with AdamW (Loshchilov & Hutter, 2017) in only 200 GRPO iterations, a small batch size of 256 samples, and without any KL regularization.

Our evaluation focuses on assessing the Conductor's capabilities to generalize across a set of challenging reasoning tasks both within and outside its training domain. For in-domain evaluation, we use all *unseen test questions* from MATH500, MMLU, RLPR, and LiveCodeBench V6. For out-of-domain evaluation, we include three *unseen test tasks* with GPQA Diamond (Rein et al., 2024), the set of diamond difficulty questions on natural science taken from the Graduate-level Google-proof Q&A benchmark, BigCodeBench (Zhuo et al., 2024), evaluating code generation and task automa-

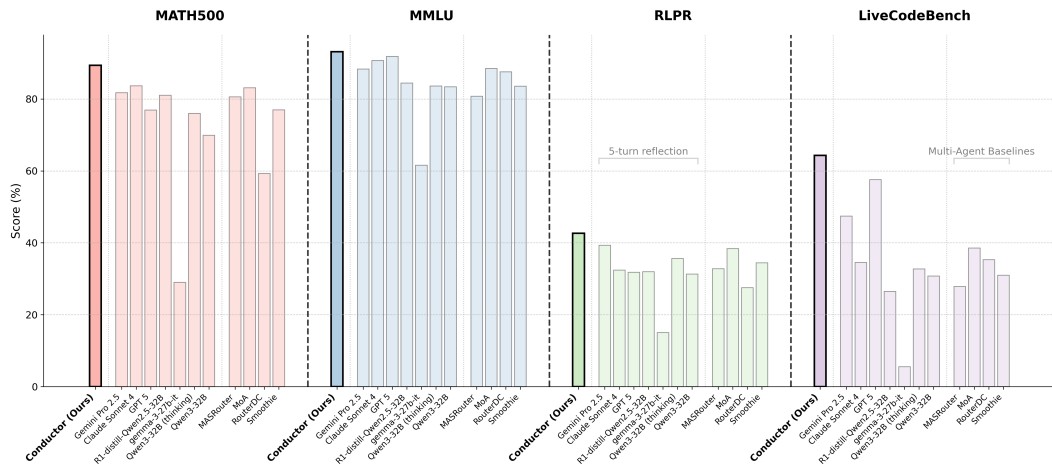

Figure 4: **Conductor in-distribution evaluation against multi-agent methods and 5-turn reflection agent baselines.** The Conductor surpasses all baselines by substantive margins, exemplifying the Conductor's ability to amplify the capabilities of its workers. Numerical results in Table 7.

tion, and AIME25 (Mathematical Association of America, 2025), the latest set of problems used for the American Invitational Mathematics Examination. These tasks cover a diverse set of competition and graduate-level models, with neither individual open nor closed-source model currently reigning supreme. To account for empirical stochasticity, we repeat each evaluation up to 16 times based on the number of questions in each task, reporting mean accuracy and standard errors. We provide all details on training and evaluation paradigms, including full hyperparameters, in Appendix A and E.

### 4.2 ELEVATING LLMS TO A NEW FRONTIER WITH THE CONDUCTOR

We first compare the Conductor's in-domain and out-of-domain performance with the previous best prior results obtained with different state-of-the-art open and closed source models. For each of our baselines, we report the "unconstrained", where all models utilize maximum reasoning and output token budgets, highest recorded performance across our own re-evaluations, private implementation, and online leaderboards, potentially including proprietary, undisclosed prompting and sampling strategies. Nonetheless, as shown in Table 1, our Conductor obtains substantial improvements compared to the very best baseline in each considered task, **attaining state-of-the-art performance records when evaluated both in-domain and out-of-domain**. For example, as shown in Fig. 1, the performance of the Conductor at the time of writing is beyond any prior LLM on the livecodebench online leaderboard[0], even surpassing the latest OpenAI's O-series models (Jaech et al., 2024), which were not included in our worker pool due to their exceedingly high cost. Furthermore, we see the Conductor, through its powerful coordination strategies, is able to generate performance gains across AIME25 and GPQA-Diamond in the range of 3%, which is consistent with entire generational improvements on these challenging benchmarks, mirroring the performance jump[12] from o3 to GPT-5, for example. We believe these results strongly validate the remarkable efficacy of our Conductor – establishing a new frontier in the capabilities of language models through collective intelligence and a new kind of powerful, adaptive agentic coordination.

### 4.3 CONTROLLED LARGE-SCALE EVALUATION

**Expensive multi-turn agentic baselines.** We also directly compare the RL Conductor with a broad range of multi-turn baselines and prior state-of-the-art adaptive routing strategies trained and evaluated with our same worker pool. First, we consider a robust and popular self-reflection agentic approach (Madaan et al., 2023b; Du et al., 2023b) where for all questions, each agent is prompted up to five times to revise and improve its answers, while keeping all its previous attempts in context. Moreover, we consider four expensive multi-agent routing coordination strategies, including MAS-Router (Yue et al., 2025), Mixture-of-Agents (MoA) (Wang et al., 2024), RouterDC (Chen et al.,

---

[0]https://livecodebench.github.io/leaderboard.html

[1]https://www.kaggle.com/benchmarks/open-benchmarks/aime-2025

[2]https://artificialanalysis.ai/evaluations/gpqa-diamond?models=o3%2Cgpt-5

2024), and Smoothie (Guha et al., 2024). In all multi-agent baselines, we train and evaluate these models with the same set of 7 agents as our Conductor, as described in Section 4.1. These prior multi-agent strategies essentially train a router classifier to construct agentic workflows by simply selecting models and/or human-designed coordination topologies from a set of pre-specified options. We note that the expressivity of these prior strategies is inherently constrained when compared to our new framework, which places complete specification freedom in the Conductor by directly using natural language as its output medium.

**Quantitative results and analysis.** We report the results from this large-scale evaluation in Figure 4, once again demonstrating how the Conductor's new unrestricted approach, with unparalleled specification freedom, yields unmatched levels of performance across each single considered task. With the sole exception of RouterDC, we note that all the baseline methods in this comparison have a strictly higher inference cost than our Conductor, which learns to construct efficient agentic workflows with an average of 3 steps, well below the requested limit despite being trained with no regularization. For example, MASRouter's performance heavily hinges on its usage of expensive human-designed agentic coordination strategies, combining 4-5 different models and roles into extensive topological

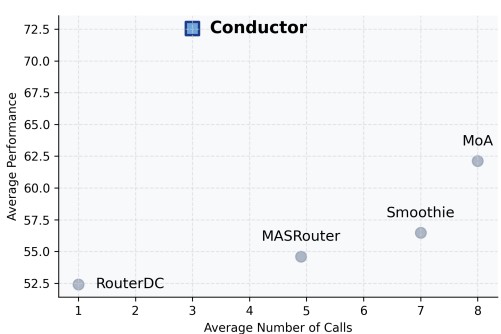

Figure 5: **Performance vs Efficiency.** The Conductor far surpasses multi-agent baselines at a fraction of the cost. Scores are task-averages from Fig. 4. Numerical results in Table 7

sequences. We believe this stark cost difference makes the Conductor's dominance even more remarkable, highlighting how our model's unrestricted prompt-engineering and task delegation capabilities provide a critical degree of adaptivity inherently beyond prior routing models and fixed human-designed strategies.

## 4.4 USER-CUSTOMIZATION AND TEST-TIME RECURSIVE SCALING

We incorporate the two extensions detailed in Section 3.2 into a pretrained Conductor with short finetuning phases. We train using a small subset of questions already seen during training, demonstrating how powerful extensions can be easily integrated without new data. We hope these experiments can provide future work with an inexpensive blueprint to adapt our powerful new model.

**Dynamic worker pool.** We evaluate our Conductor finetuned on randomized model subsets and compare it with its pre-trained counterpart, which was always given full access to all models in our original set. In particular, we focus on two evaluation user cases, restricting our models to use either exclusively the closed or open-source subsets of models. We note that before our targeted finetuning phase, the Conductor relied on open-source models only in very specific scenarios, given their significantly inferior performance for most tasks. Nonetheless, as shown in Figure 6, when evaluated with only open models, the finetuned Conductor is able to effectively combine their individually weaker capabilities with surprising effectiveness, even consistently outperforming Claude Sonnet 4 by almost 10% within our constrained setting. This demonstrates a core capability of the Conductor, which is that the Conductor is

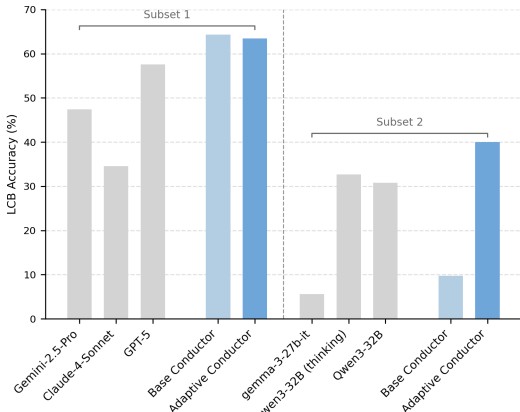

Figure 6: Finetuned on randomized model pools, the Conductor achieves strong performance over rarely used open-model subsets while maintaining performance on the closed-model subsets.

not exclusively reliant on the performance foundation of frontier models, and indeed displays even larger absolute gains when using a foundation with a larger room for improvement. At the same time, when evaluated with only closed models, the finetuned Conductor does not compromise its original

Table 2: **Test-time recursion generates further performance gains.** To accommodate recursion, we use our controlled evaluation setting described in Section 4.3. When allowed to specify itself as a worker LLM and adaptively revise its initial coordination strategy at test-time, the Conductor unlocks substantive additional gains on BigCodeBench.

| Model | AIME25 | BigCodeBench | GPQA-D | Average score |
|---|---|---|---|---|
| gemma-3-27b-it | 6.67 | 10.8 | 33.33 | 16.93 |
| Qwen3-32B | 23.33 | 23.0 | 54.05 | 33.46 |
| Qwen3-32B (thinking) | 23.33 | 20.9 | 59.09 | 34.44 |
| R1-Distill-Qwen-32B | 30.00 | 24.3 | 51.01 | 35.10 |
| Gemini Pro 2.5 | 46.67 | 35.1 | 75.25 | 52.34 |
| Claude Sonnet 4 | 35.33 | 35.8 | 67.30 | 46.14 |
| GPT 5 | 46.67 | 33.8 | 72.73 | 51.73 |
| *Conductor* (**Ours**) | **66.67** | 37.8 | 81.31 | **61.93** |
| *Conductor-Recursive* (**Ours**) | **66.67** | **40.0** | **82.32** | **63.00** |

Figure 7: **Conductor Scale.** The 3B Conductor still learns optimal agent selection, as shown by the agent distribution converging on the three most powerful models (left). However, when scaling to 7B, the Conductor generates additional performance gains, even for identical agent selection, through its improved prompt engineering (right). Evaluation performance taken from Fig. 4.

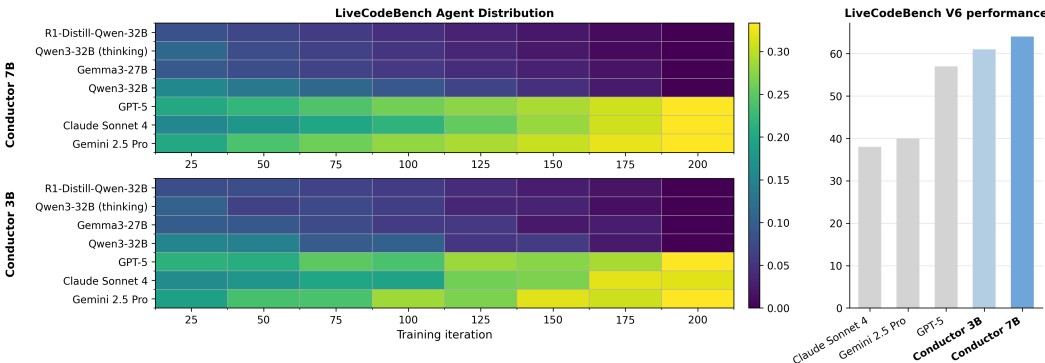

state-of-the-art behavior, entirely matching its pretrained performance. These results demonstrate that our new model can be readily extended to cater to individual user requirements and help mitigate the field's inherent cost-performance tradeoffs.

**Self-referential recursive scaling.** We evaluate our finetuned recursive Conductor on out-of-distribution tasks, where both our pretrained model and all its individual workers still show potential for improvement. In particular, we observe that some generally proficient coding models, such as GPT5, behave suboptimally when evaluated on BigCodeBench, requiring agentic workflows significantly differing from those learned during pretraining. We present our results in Table 2, showing how incorporating recursion yields a marked performance boost, especially evident on this challenging benchmark. We note that we set our recursive variant to use less than $2\times$ the number of its original agentic calls to mitigate additional costs, potentially leaving room for further improvement on the Conductor's performance by simply increasing compute. Delving deeper into our results, as shown in Figure 10, we find the Conductor effectively redistributes its agent selection towards Claude 4 and Gemini 2.5 during its BigCodeBench recursive calls after observing the unexpectedly suboptimal behavior of GPT5. Overall, these results compellingly demonstrate the recursive Conductor's newfound ability to intelligently design and adjust its agentic workflows on the fly, concretely improving its effectiveness and robustness to unseen test scenarios.

### 4.5 ANALYZING AND ABLATING THE PROPERTIES OF AN EFFECTIVE CONDUCTOR

**Conductor scale.** We start our analysis of the Conductor's remarkable capabilities by investigating the role played by model scale. To this end, we examine and compare the behavior of our 7B Conductor by training a smaller variant with 3B parameters following the same recipe. We then analyze how the agentic workflows designed by the two models differ, together with their final performance on LiveCodeBench. As shown in Figure 7, as training progresses, we find that the two Conductors

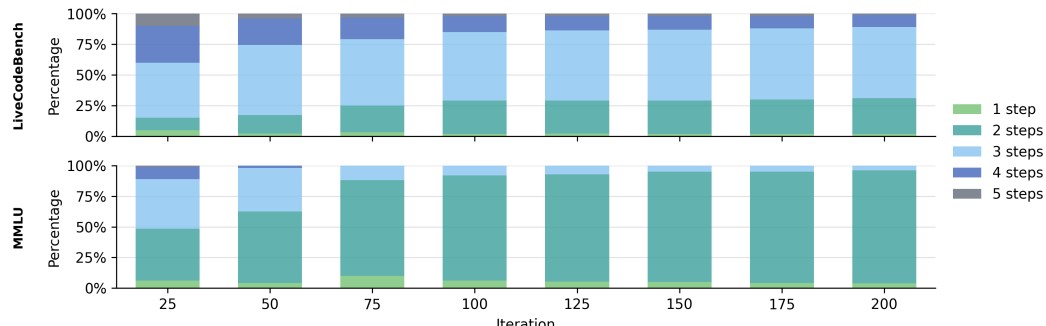

Figure 8: **Task adaptivity.** In more straightforward tasks, such as MMLU, the Conductor learns that 2 agents working together is optimal. In more complex settings, such as LiveCodeBench, the Conductor allocates more compute by devising coordination strategies with 3 or even 4 agents.

converge to select the same distribution of worker agents. However, while both of our models still display performance well beyond all of our baselines, our larger 7B variant maintains a clear edge at the end of training. Comparing the substasks specified by the 7B Conductor and its 3B counterpart, illustrated respectively in Figures 3) and 15 of Appendix F, we trace this performance gap to the larger model's superior *prompt engineering* skills. This relationship further highlights how our new framework opens a new axis for scaling multi-agent coordination far beyond prior routing efforts. Together with our results, we believe this analysis further evidences the importance of removing manual constraints on subtask specification: enabling the increased natural language capabilities of larger and newer base models to directly translate into more intelligent prompt engineering and allowing the Conductor to unlock a new level of agency over each of its workers.

**Task and difficulty adaptivity.** Additionally, we study how distinct input tasks of varying difficulty influence our Conductor. Our analysis reveals a compelling *emergent* behavior, with the model learning to dynamically allocate more compute to harder problems by specifying agentic workflows with an increased number of steps. We illustrate this phenomenon in Figure 8, showing a stark difference between the distributions of workflow steps for LiveCodeBench code generation and simpler MMLU multiple-choice questions. As training progresses, the Conductor learns to divide complex LiveCodeBench problems into increasingly granular subtasks, often deploying multiple planning steps followed by implementation and verification attempts. In contrast, as MMLU primarily tests factual knowledge and comprehension, the Conductor produces simpler workflows, typically limited to one or two steps of targeted information retrieval. By inspecting the output traces presented in Appendix F, we also observe that the model explicitly reasons about task complexity before specifying its workflows. Coupled with its notable efficiency, this adaptivity highlights how natural language grounding is again a critical factor in our model's adaptivity, enabling workflows that dynamically match problem difficulty while avoiding wasted computation on simpler cases.

## 5 RELATED WORK

**Reinforcement learning with tools.** Reinforcement learning (RL) has become an increasingly popular paradigm for the elicitation of reasoning capabilities in LLMs (Guo et al., 2025; Lambert et al., 2024; Chu et al., 2025). Recent works aim to extend the RL paradigm to go beyond pure textual reasoning through tool use, enhancing capabilities in geometric reasoning, complex equation solving, code execution, search-augmented question answering, and precise computation (Gehring et al., 2024; Feng et al., 2025; Yu et al., 2024; Le et al., 2022; Nakano et al., 2021). For example, Gehring et al. (2024) and Le et al. (2022) incorporate execution feedback on code synthesis and unit tests into end-to-end RL, Feng et al. (2025) use dynamic interleaving of real-time code execution, while Nakano et al. (2021) equip the base model with a text-based web-browser. Yu et al. (2024) consider multi-step tool usage through step-grained reward shaping. Our framework establishes a new extension to the tool-using RL paradigm, where powerful collaborative reasoning topologies emerge from RL by equipping the base model with workflow delegation through API calling.

**Multi-agent coordination.** With increasingly powerful individual LLMs, recent works aim to design topological and prompt-based scaffolds to coordinate groups of agents (Du et al., 2023a; Wang et al., 2024; Dang et al., 2025; Madaan et al., 2023a; Yue et al., 2025). Wang et al. (2024) and Du et al. (2023a) propose carefully hand-designed scaffolds to orchestrate agents within and across successive rounds, Guha et al. (2024) and Yue et al. (2025) learn embedding spaces to map queries to agents and topologies, Zhuge et al. (2024) treat collaboration as a learnable graph, and Chen et al. (2024) learn a router to direct queries to the single best-matched agent. Our framework differs from all existing approaches by learning powerful agent coordination strategies through pure end-to-end RL, allowing the Conductor complete freedom to learn any strategy expressible in natural language.

## 6 DISCUSSION AND EXTENSIONS

In this work, we introduce the Conductor, a new language model trained with reinforcement learning to push the boundaries of frontier LLMs through collective intelligence and automated prompt engineering. By dividing challenging problems, delegating targeted subtasks, and designing effective communication topologies, a 7B Conductor attains state-of-the-art performance across a diverse set of highly competitive benchmarks, going well beyond manually designed agentic pipelines and expensive multi-agent baselines. Furthermore, we demonstrate that the capabilities of our pretrained Conductor can be easily extended via finetuning, giving users the ability to specify customized agent sets and unlocking a new form of test-time scaling with recursive calling to the Conductor itself. By sharing our full code with this submission, we hope to facilitate future efforts in using language models themselves as intelligent *meta-agents*, flexibly harnessing the complementary capabilities of a broader set of models. To this end, an exciting, unexplored extension is to go beyond LLMs alone, introducing workers with expertise in other modalities (Jumper et al., 2021; Intelligence et al., 2025), allowing the Conductor to use natural language as an expressive unifying interface and tackle increasingly ambitious human challenges in fields such as biology, robotics, and beyond.

## ACKNOWLEDGEMENTS

We thank Koshi Eguchi and Kou Misaki for the infrastructure support, and the entire Sakana AI R&D team for their valuable comments and suggestions.

## AUTHORS CONTRIBUTIONS

Stefan Nielsen proposed the Conductor as an LLM that reasons over collaboration topologies and subtasks, and implemented and tuned the Conductor. Edoardo Cetin proposed the Conductor as an LLM that reasons over collaboration topologies and subtasks and implemented the core algorithm and training paradigm. Peter Schwendeman proposed and implemented essential prompting and training improvements, tuned the Conductor, built the baseline suite, and conducted analyses that strengthened and validated the system. Qi Sun implemented and conducted the out-distribution task evaluation. Jinglue Xu designed and curated the training datasets. Yujin Tang initiated and led the project, implemented the initial algorithm, and conducted the first experiments. All authors contributed to the experimental design and paper writing.

**Reproducibility statement.** We provide source code in the supplementary material along with our trained model weights in an anonymized external repository. We additionally provide the full details of our experimental setup – including datasets, model specification, training regime, and evaluation protocol – in Appendix A and E. Our base model and all datasets are publicly available.

**Ethics statement.** Our work considers multi-agent collaboration, and in particular, the connection between reinforcement learning, reasoning, and the capabilities of small LLMs to automatically discover optimal coordination strategies over LLM workers. Given this, we see foresee no issues regarding fairness, privacy, or security, or any other harmful societal or ethical implications outside broader considerations for the field itself. However, we do note that the reliance of our method on expensive language models might further exacerbate the economic divide and barriers posed by AI.

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

Table 3: **Completion tokens and reasoning budgets in the unconstrained setting**

| Model | Max Completion Tokens | Reasoning Budget |
|---|---|---|
| Gemini Pro 2.5 | 65535 | 32768 |
| Claude Sonnet 4 | 64000 | 32768 |
| GPT 5 | 128000 | *high* |
| R1-Distill-Qwen-32B | 20480 | *N.A* |
| gemma-3-27b-it | 20480 | *N.A* |
| Qwen3-32B (thinking) | 20480 | *enabled* |
| Qwen3-32B | 20480 | *N.A* |

## A  EXPERIMENTAL DETAILS

### A.1  TRAINING SETUP

**Model, optimizer, and training regime.** We use Qwen2.5-7B (Hui et al., 2024) as our base model with a max completion length to 1024. We train for 200 iterations, sampling 4 questions per iteration and generating 64 rollouts per question with a temperature of 1.0. We use AdamW (Loshchilov & Hutter, 2017) as optimizer with $\beta_1 = 0.9$, $\beta_2 = 0.999$, $\epsilon = 0.2$ and a base learning rate of 0.000001 with cosine scheduling and a warmup ratio of 0.03. We disable reference model synchronization and set the reference model KL divergence penalty to 0.

**Agent settings.** We set all agent workers to 4096 max completion tokens and decode with a temperature of 0.2. Reasoning budgets for the closed source models are set to their minima, which is 128 tokens for Gemini 2.5 Pro, 0 for Claude Sonnet 4, and 'minimal' for GPT-5. For Qwen3-32B (both non-thinking and thinking modes), we set their decoding parameters as top-p=0.8, top-k=20, and use a presence penalty of 1.0.

**Recursion.** We train our Conductor-Recursion by taking our trained Conductor and finetuning for 20 iterations on a 350 sample filtered subset of our training dataset, comprising 175 LiveCodeBench and 175 RLPR questions. We continue with the same configuration of 64 rollouts per sample, amounting to batch size of 256, and use no reference model synchronization or KL divergence penalty. We use a discount factor of 0.25 to scale the rewards in the initial, non-recursive round and normalize rewards across rounds.

**Compute resources.** We train our Conductor on 2 NVIDIA H100 80GB GPUs.

### A.2  DATASETS

We select the following tasks for training and evaluation both for their usage in other multi-agent works and for their difficulty and popularity in measuring frontier model performance. For instance, MATH500 is used in (Yue et al., 2025) and MMLU is used in (Yue et al., 2025; Chen et al., 2024). We also note that we chose harder versions of popular benchmarks in multi-agent papers. For example, Yue et al. (2025); Chen et al. (2024); Du et al. (2023b) use GSM8K to evaluate mathematical reasoning, and we chose the more challenging AIME25.

**MMLU** (Hendrycks et al., 2020). A massive multitask language comprehension dataset spanning 57 tasks, including history, law, sciences, and social sciences, among others, testing world knowledge and problem solving ability. All questions are multiple choice. The training data comprises 99842 samples and the test data comprises 14042 samples.

**MATH500** (Lightman et al., 2023). A subset of 500 of the most challenging problems selected by Lightman et al. (2023) from the MATH dataset (Hendrycks et al., 2021). We construct a train set of 300 samples and a test set of 100 samples.

**LiveCodeBench** (Jain et al., 2024). This is the coding subset of the Live Bench (White et al., 2024), which is a contamination-free, continuously updated benchmark spanning 17 diverse tasks including coding, math, data analysis, and language among others. LiveCodeBench comprises leetcode-style code generation problems. We select the oldest version, version 1, as our training data, and the newest version (at the time of writing), version 6, for testing.

**RLPR** (Yu et al., 2025). A non-mathematical, general reasoning dataset obtained as a subset from WebInstruct (Yue et al., 2024), where samples are filtered using GPT-4.1 to remove overly easy questions. The total dataset comprises 77700 samples, of which we take 46620 as training data and 15540 as test data.

**AIME25** (Mathematical Association of America, 2025). One of the most challenging and popular mathematical reasoning benchmarks, this dataset is the 30 questions used in the 2025 edition of the annual American Invitational Mathematics Examination. We use all 30 questions for evaluation.

**GPQA-Diamond** (Rein et al., 2024). This is the set of diamond difficulty problems on natural science taken from the Graduate-level Google-proof Q&A benchmark. The dataset comprises 198 problems in multiple-choice format. We use all 198 for evaluation.

**BigCodeBench** (Zhuo et al., 2024). A challenging code generation benchmark focusing on diverse function calls and challenging instructions. We select the "hard" and "complete" subset, comprising 148 samples, which is specifically designed for code completion based on comprehensive docstrings. We use all 148 samples for evaluation. For all code evaluation we use the default gradio backend, hosted on the HuggingFace space[3].

## A.3 EVALUATION SETUP

**Unconstrained setting.** We refer to the setting in which we set the completion tokens and reasoning budget limits to their max as *unconstrained*. This is the setting with which we obtain our results in Section 4.2. Completion tokens and reasoning budgets are detailed in Table 3. In the case of Qwen3-32B (thinking), no budget is configurable, instead we simply select "enabled". We make one exception to the max reasoning configuration in the unconstrained evaluation setup, which is for GPT-5 in BigCodeBench, where we evaluate GPT-5 at medium reasoning effort as opposed to high reasoning effort. This is due to GPT-5's performance being marginally stronger under medium reasoning effort, which is in line with OpenAI's own findings regarding GPT-5 (OpenAI, 2025), in which for certain tasks medium may outperform high reasoning effort.

**Constrained setting.** For our evaluation in Section 4.3, we evaluate the Conductor under cost constraints (which we term *constrained*) in which all agent models are capped to 4096 output tokens and all reasoning budgets are set to their minima. The reasoning budget minima are "minimal", 128, and 0 and "disabled" for GPT 5, Gemini 2.5 Pro, Claude Sonnet 4, and Qwen3-32B (thinking) respectively. Note this is the identical setting with which we train the Conductor.

We evaluate the Conductor with recursion (Section 3.2) using this same constrained setting to accommodate the fact the Conductor will be passed the final worker response as part of its context.

We evaluate AIME25 and GPQA-Diamond using Lighteval (Fourrier et al.) and BigCodeBench using the original source repository[4].

**Baselines.** Within the constrained setting, we evaluate against state-of-the-art multi-agent routing and scaffolding techniques, including MasRouter (Yue et al., 2025), RouterDC (Chen et al., 2024), Smoothie (Guha et al., 2024), and MoA (Wang et al., 2024) as seen in Figure 4 and Table 7. We train MasRouter and RouterDC using the by sampling from the training dataset that was used to train Conductor. We select the evaluation model based on best validation loss from a validation set which is of the same distribution, but independent of both the training and testing sets. Specifically, MasRouter is trained with a batch size 256 and validated every 5 iterations. We stop training early after seeing sufficient evidence of overfitting (i.e. 10% drop off in validation). RouterDC is trained on 500 samples, with each sample repeated 5 times to collect an average performance of every work on the given question. The collected average performance is used to compute the contrastive loss described in Chen et al. (2024). Smoothie is applied to test-time questions and outputs of each worker model with both dependent (selecting a model per question) and independent (selecting a single model for all questions) strategies. MoA is also applied as a test-time scaffold with a single MoA layer and single aggregator layer for a total of 8 model calls. The aggregator model is chosen at random. All baselines were compared against using the code and default settings provided by their respective authors.

---

[3]https://huggingface.co/spaces/bigcode/bigcodebench-evaluator
[4]https://github.com/bigcode-project/bigcodebench

Table 4: OOD few-shot prompting boosts performance. Conductor performance is increasing in the proportion of few-shot examples it's provided with that are taken from unseen tasks.

| Model | MATH500 | MMLU | RLPR | LiveCodeBench |
|---|---|---|---|---|
| In-distribution | 88.20 | 92.31 | 42.60 | 58.32 |
| Mixed OOD and In-distribution | 88.70 | 92.62 | 42.60 | 61.43 |
| OOD (**Ours**) | **89.33** | **93.14** | **42.63** | **64.29** |

In all tasks, we report the best of either our own implementation or any existing online leaderboards that use the same configuration. In cases where we use an online leaderboard's reported score for any model, we then match that score's precision, hence why we report, for example, AIME25 and GPQA Diamond to 1 decimal place and BigCodeBench to 2 decimal places.

**Performance improvement scale.** We note that in our evaluation setup we focus on highly competitive reasoning benchmarks, such as AIME and GPQA-Diamond. Such tasks tend to have a long-tailed distribution of difficulty (Xu et al., 2025) where breakthroughs in a small subset of particularly challenging problems could be representative of entire generational improvements in LLM reasoning. For example, the difference between two generations of GPT reasoning models, from GPT-o3 and GPT-5, is 3.3% on AIME25[5] and 2.7% on GPQA-Diamond[6] in absolute percentage terms. Moreover, the Conductor framework not only yields improvements in a single domain but across a diverse range of such highly challenging benchmarks, across math, coding, and natural science, something we believe particularly validates the meaningfulness of its advancements beyond the performance foundation of frontier models.

# B    ADDITIONAL EXPERIMENTAL RESULTS

## B.1    EFFICIENCY ANALYSIS

We present in Table 5 additional efficiency results for the Conductor in comparison with Claude Sonnet 4, Gemini 2.5 Pro, and GPT-5 using a consensus inference-time scaling framework (Wang et al., 2022). We take MMLU as a representative task, set the consensus sampling to 5 (in keeping with the Conductor's max allowable workflows of 5 steps), and report average token usage per sample, average cost per sample, and cost adjusted performance (taken simply as performance / average cost per sample in cents). We see the Conductor not only outperforms consensus, but additionally offers substantial efficiency gains in terms of token usage and average cost relative to this popular inference-time scaling technique.

Table 5: Efficiency comparison with 5× inference-time scaling (consensus vs. reflect).

| Model | Performance | Token Usage | Avg. Cost | Cost-adjusted Performance |
|---|---|---|---|---|
| Claude 5× consensus | 91.00 | 1412.8 | 0.0211 | 42.94 |
| Claude 5× reflect | 90.66 | 2517.0 | 0.0208 | 43.58 |
| Gemini 5× consensus | 91.60 | 1658.4 | 0.01658 | 55.23 |
| Gemini 5× reflect | 88.33 | 2919.8 | 0.01675 | 52.70 |
| GPT 5 5× consensus | 91.30 | 1376.3 | 0.0138 | 66.34 |
| GPT 5 5× reflect | 91.79 | 2457.132 | 0.0142 | 64.42 |
| Conductor | **93.14** | **735.2** | **0.009** | **103.49** |

We present in Table 6 additional efficiency results for the Conductor in comparison with multi-agent baselines. We report average performance, token usage per sample, and average cost per sample, with averages taken over the four-way mixed training dataset of MMLU, RLPR, LiveCodeBench, and MATH500. We see that, in line with Figure 5, the low API calls of the Conductor translates

---

[5]Kaggle leadboard: https://www.kaggle.com/benchmarks/open-benchmarks/aime-2025
[6]Artificial Analysis leaderboard: https://shorturl.at/eLHUj

into substantive efficiency gains, attaining second lowest token usage and highly competitive cost efficiency while outperforming all other baselines by large margins.

Table 6: Efficiency comparison across multi-agent baselines.

| Model | Performance | Token Usage | Cost |
|---|---|---|---|
| MoA | 62.13 | 11203 | 0.04855 |
| Smoothie | 56.48 | 9909 | 0.03929 |
| RDC | 52.41 | **840** | **0.00561** |
| MasRouter | 56.89 | 4970 | 0.01345 |
| Conductor | **72.35** | 1820 | 0.02384 |

### B.2 Inferring agent compatibility through OOD few-shot prompting

As described in Section 3, we supply the Conductor with few-shot examples of known, successful coordination strategies in order to condition the generative distribution of the pretrained language model to the orchestration task at hand, raising the probability of properly formatted completions at initialization.

We find, perhaps surprisingly, that Conductor performance is *increasing* in the proportion of few-shot examples taken from OOD tasks, where best performance is attained when all few-shot examples are OOD. For example, we find that if training the Conductor to solve coding problems, providing the Conductor with successful coordination strategies on non-coding tasks, such as math problems, boosts performance on coding tasks, even outperforming a setting in which the Conductor is provided with successful coordination strategies on coding problems. This finding is empirically demonstrated in Figure 9 and Table 4, where we see clear separation throughout training and at final evaluation according to the proportion of OOD few-shot examples.

We posit that this behavior arises due to the fact that the tasks, by being OOD, prevent the Conductor from exploitation of the provided strategies and better incentivize exploration of the coordination strategy space. In this sense, the OOD few-shot examples help to deliver useful information regarding compatible combinations of agents, but isolate this information from a reward-hackable strategy that can be lazily repeated.

### B.3 Performance Diversity

Throughout our evaluation, we find, in line with existing leaderboards and surveys (Chang et al., 2024), that no single model reigns supreme over all tasks, with differing models excelling or struggling in differing tasks. Examples of this in our own evaluation include GPT-5's strong performance in math and competitive coding (seen in AIME and LiveCodeBench), while Gemini excels in scientific reasoning (GPQA-Diamond). Claude Sonnet 4 struggles at competitive coding (relatively weak in LiveCodeBench), but is one of the dominant models at code generation with diverse function calling (BigCodeBench). This specialization is also prevalent at the granular "sub-task" level, where we often find the Conductor learns that different models are most useful as "planners" or "writers" to answer particular kinds of questions. For instance, our SOTA performance in LiveCodeBench leverages Gemini 2.5 Pro and Claude Sonnet 4 working together as high-level planners and only later employs GPT-5 to write the final optimized code, which far outperformed alternate strategies using GPT-5 in a planning role that the Conductor attempted at early training iterations.

Furthermore, regarding 'weaker' open-source models, we even observed concrete instances where these models could fill roles and solve particular questions that their closed-source counterparts failed at. We note this was more prevalent at a subtask level, where using GPT5 as the final validator for several BigCodeBench questions would fail to adhere to the benchmark's strict formatting requirements, disregarded by this agent. In these instances, simply switching away to the much smaller Qwen3-32 or DeepSeek as final validators would allow the Conductor to succeed. While expectedly less common, we also found examples at the global "task level", with most such cases in RLPR and MMLU.

Table 7: **Self-reflection and multi-agent baseline comparison.** The Conductor outperforms all multi-agent baselines and all individual worker agents, including when evaluated at 5× context length and 5× self-reflection.

| Model | MATH500 | MMLU | RLPR | LiveCodeBench | Avg. |
|---|---|---|---|---|---|
| Gemini Pro 2.5 (4K/128) | $85.30 \pm 1.42$ | $91.53 \pm 0.26$ | $39.57 \pm 1.50$ | $40.14 \pm 2.20$ | 64.14 |
| Claude Sonnet 4 | $82.90 \pm 1.59$ | $90.66 \pm 1.01$ | $32.60 \pm 0.35$ | $38.00 \pm 1.50$ | 61.04 |
| GPT 5 (4K/minimal) | $74.45 \pm 2.19$ | $89.79 \pm 0.65$ | $33.13 \pm 1.29$ | $57.50 \pm 2.32$ | 63.72 |
| DeepSeek-R1-Distill-Qwen-32B | $78.50 \pm 1.99$ | $84.41 \pm 0.87$ | $32.75 \pm 1.56$ | $24.86 \pm 0.90$ | 48.95 |
| gemma-3-27b-it | $37.45 \pm 7.84$ | $63.58 \pm 2.26$ | $14.93 \pm 4.99$ | $7.21 \pm 2.07$ | 30.79 |
| Qwen3-32B (reasoning) | $76.85 \pm 1.79$ | $83.28 \pm 0.20$ | $34.35 \pm 0.98$ | $31.21 \pm 2.16$ | 56.42 |
| Qwen3-32B (direct) | $73.15 \pm 2.25$ | $84.02 \pm 0.56$ | $30.60 \pm 0.82$ | $26.79 \pm 1.48$ | 53.64 |
| *5× Context Length* | | | | | |
| Gemini Pro 2.5 (20K/128) | $86.40 \pm 1.39$ | $91.51 \pm 0.24$ | $39.57 \pm 1.50$ | $52.93 \pm 2.16$ | 67.60 |
| Claude Sonnet 4 | $82.20 \pm 1.54$ | $86.93 \pm 0.54$ | $32.42 \pm 0.81$ | $37.93 \pm 1.18$ | 59.87 |
| GPT 5 (20K/minimal) | $75.50 \pm 2.89$ | $89.42 \pm 0.34$ | $32.68 \pm 1.09$ | $58.36 \pm 2.15$ | 63.99 |
| DeepSeek-R1-Distill-Qwen-32B | $82.50 \pm 1.76$ | $84.43 \pm 0.64$ | $33.50 \pm 0.78$ | $26.86 \pm 0.33$ | 50.11 |
| gemma-3-27b-it | $39.80 \pm 8.16$ | $81.28 \pm 0.14$ | $16.67 \pm 2.70$ | $13.14 \pm 2.09$ | 37.72 |
| Qwen3-32B (reasoning) | $76.85 \pm 1.79$ | $84.08 \pm 0.36$ | $34.35 \pm 0.98$ | $25.86 \pm 1.25$ | 55.29 |
| Qwen3-32B (direct) | $73.50 \pm 2.14$ | $83.54 \pm 0.40$ | $31.00 \pm 0.85$ | $21.21 \pm 1.60$ | 52.31 |
| *5× Self-Reflection* | | | | | |
| Gemini Pro 2.5 | $81.75 \pm 1.80$ | $88.33 \pm 0.37$ | $39.30 \pm 1.99$ | $47.43 \pm 1.67$ | 64.20 |
| Claude Sonnet 4 | $83.66 \pm 1.74$ | $90.66 \pm 0.74$ | $32.42 \pm 0.81$ | $34.56 \pm 0.81$ | 60.33 |
| GPT 5 | $76.93 \pm 2.40$ | $91.79 \pm 0.07$ | $31.80 \pm 2.00$ | $57.57 \pm 2.07$ | 64.52 |
| DeepSeek-R1-Distill-Qwen-32B | $81.00 \pm 1.73$ | $84.41 \pm 0.15$ | $32.32 \pm 0.36$ | $26.50 \pm 0.75$ | 49.48 |
| gemma-3-27b-it | $29.00 \pm 5.94$ | $61.57 \pm 0.56$ | $15.05 \pm 6.21$ | $5.57 \pm 0.90$ | 27.80 |
| Qwen3-32B (reasoning) | $76.00 \pm 2.65$ | $83.60 \pm 0.51$ | $35.90 \pm 0.26$ | $32.71 \pm 2.30$ | 57.05 |
| Qwen3-32B (direct) | $69.90 \pm 2.95$ | $83.37 \pm 0.16$ | $31.33 \pm 0.32$ | $30.79 \pm 1.54$ | 53.85 |
| *Scaffolding / Aggregation Baselines* | | | | | |
| MASRouter | $80.60 \pm 0.89$ | $86.28 \pm 2.77$ | $32.80 \pm 4.77$ | $27.86 \pm 3.24$ | 56.89 |
| MoA | $83.10 \pm 2.65$ | $88.46 \pm 0.76$ | $38.37 \pm 0.95$ | $38.57 \pm 3.50$ | 62.13 |
| RouterDC | $59.25 \pm 4.22$ | $87.52 \pm 0.06$ | $27.53 \pm 2.22$ | $35.33 \pm 2.34$ | 52.41 |
| Smoothie (Independent) | $76.85 \pm 1.74$ | $83.28 \pm 0.16$ | $34.35 \pm 0.80$ | $31.21 \pm 2.02$ | 56.42 |
| Smoothie (Dependent) | $76.95 \pm 2.06$ | $83.56 \pm 0.27$ | $34.45 \pm 0.67$ | $31.00 \pm 2.04$ | 56.48 |
| *Conductor* (**Ours**) | $\mathbf{89.33} \pm 0.58$ | $\mathbf{93.14} \pm 0.36$ | $\mathbf{42.63} \pm 0.65$ | $\mathbf{64.29} \pm 2.01$ | **72.35** |

## B.4 Controlled large scale evaluation full results

We present in Table 7 the full numerical results for our controlled large scale evaluation across all worker models and multi-agent baselines. For Smoothie, we train both a independent and dependent versions, where the independent version selects a single model for all questions and the dependent versions selects a different model depending on the specific question. We report the best performing Smoothie variant in our Figure 4 in the main text in Section 4.

## B.5 Large scale evaluation extended discussion

We note that on LiveCodeBench, the MoA baseline underperformed GPT-5, failing to leverage GPT-5's impressive capabilities. Examining the evaluation logs reveals the drop in performance is explained by MoA suboptimally using the candidate solutions of less capable models to inform the final response, or often being misled by the incorrect solutions of other models. Indeed, we see that for other tasks where models are closer in capability, MoA performed better, but for Live-CodeBench where there is a high variance in performance, MoA struggled to discern the correct answer or combination of answers among the 7 candidate responses. We note that this finding echoes the result obtained in Table 11, showing that open-weight models can degrade the performance of the frontier models when combined suboptimally, with particularly marked performance drops in

Table 8: **Out-of-Distribution evaluation under cost constraints.** The Conductor continues to deliver performance gains when zero-shot transferred to new, unseen tasks.

| Model | AIME25 | BigCodeBench | GPQA-D | Avg. |
|---|---|---|---|---|
| R1-Distill-Qwen-32B | 30.00 | 24.3 | 51.01 | 35.10 |
| gemma-3-27b-it | 6.67 | 10.8 | 33.33 | 16.93 |
| Qwen3-32B (thinking) | 23.33 | 20.9 | 59.09 | 34.44 |
| Qwen3-32B | 23.33 | 23.0 | 54.05 | 33.46 |
| Gemini Pro 2.5 | 46.67 | 35.1 | 75.25 | 52.34 |
| Claude Sonnet 4 | 35.33 | 35.8 | 67.30 | 46.14 |
| GPT 5 | 46.67 | 33.8 | 72.73 | 51.07 |
| *Conductor* (**Ours**) | **66.67** | **37.8** | **81.31** | **61.93** |

LiveCodeBench. We posit that this reveals a property of MoA, for which performance depends in large part on the ability to discern correct from incorrect, which becomes increasingly challenging in tasks with very large solution spaces (for example writing optimized code).

One concrete example illustrating this point is the following example taken from MMLU. Here, the open-weight models produced incorrect reasoning and responses, ultimately leading to the incorrect selection: *Many years ago, children who had good manners kept quiet if their parents were talking with other persons. Today, well-mannered children have more freedom. Sometimes good manners in one place are bad manners in other place...[truncated]...Which of the following sentences is not true according to the passage? Options: A. Well-mannered children should always keep quiet.B. Eating with others is bad manners. C. Good manners are different from one place to another.D. People always want others to bother them.* In this example, Gemini 2.5 Pro correctly identified C as the answer but the final response was misinformed by the erroneous reasoning of other responses.

Regarding MASRouter, we instead find that this baseline relies heavily on human-engineered scaffolding techniques which require careful placements of selected models for specific roles. Therefore, when asked to make many specific decisions about model and role selection, MASRouter this framework can struggle to determine which models are best suited for each task. This issue is especially evident when MASRouter is exposed to our wide pool of worker agents, where discovering the optimal combinations of agents and allocations of roles is challenging. Additionally, the manually designed setup of MASRouter forces it to rely on fixed prompt templates and scaffolds to direct models to solve problems, with limited generality beyond the domains it was designed for. By contrast, the Conductor uses no human-designed fixed prompts, can leverage LLM's strong generalization properties and learn how to write custom focused subtasks that best leverage their capabilities for each question.

### B.6 ZERO-SHOT GENERALIZATION AT BOUNDED CONTEXT

We additionally provide experimental results for our Conductor when zero-shot transferred to unseen tasks and evaluated under the const constrained setting described in Section 4.3. We see marked performance gains across all OOD tasks in this setting, mirroring our findings for the in-distribution setting presented in Section 4.3. We note in BigCodeBench the somewhat surprising result that Qwen3-32B outperforms Qwen3-32B (thinking). Analyzing the completion transcripts, we see that this performance decrease typically stems from added verbosity causing formatting failures. Indeed, this is similar to the situation observed in GPT-5, where medium reasoning effort outperformed high reasoning effort.

### B.7 ABLATION STUDIES

We ablate the subtasks by retraining the Conductor with an alternate prompt, identical to 13, but with the requirement to generate subtasks removed. Instead, all models selected in the coordination strategy are uniformly prompted with 'Solve the user question'. The Conductor is thereby trained only to work out optimal agent combinations and collaboration topologies.

Table 9: **Ablations studies on subtasks, few-shot conditioning, and utilizing fine-grained complex topology specification**

| Model | MATH500 | MMLU | RLPR | LiveCodeBench |
|---|---|---|---|---|
| fine-grained | 88.67 | **93.55** | 42.28 | 61.24 |
| w/o few-shot | 82.00 | 92.69 | 41.50 | 54.86 |
| w/o subtasks | 88.5 | 92.75 | 41.95 | 58.62 |
| Conductor (**Ours**) | **89.33** | 93.14 | **42.63** | **64.29** |

**Subtasks.** We present results of the ablation in Table 9, where the ablation model is denoted *w/o subtasks*. We see a consistent drop in performance across all tasks, with a particularly substantial drop in LiveCodeBench. This result reveals that the necessity of careful and targeted prompt engineering is increasing in overall task complexity and, in particular, instruction complexity. Solving LiveCodeBench code-generation problems successfully typically require abiding by numerous constraints and formatting requirements, while ensuring code is accurate and runable. By contrast, MATH500, MMLU, and RLPR questions are typically just a few sentences, meaning direct requests to solve perform reasonably well.

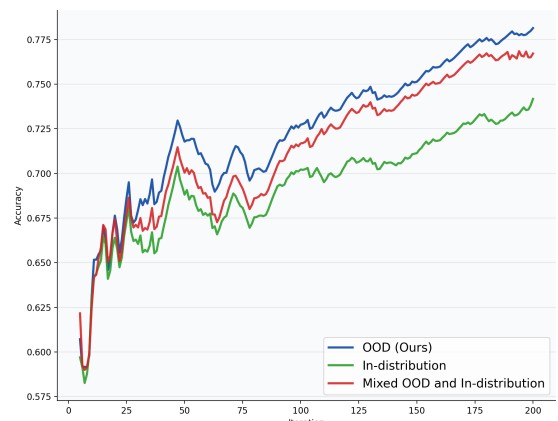

Figure 9: **OOD few-shot examples improve Conductor performance.**

This therefore reveals a promising scope of application for the Conductor, namely that for increasingly difficult and task complexity, in which instruction following and abiding by constraints is of utmost importance, the performance gains accessible by the Conductor increase as well.

**Agent selection.** We ablate the agent selection produced by the Conductor by fixing all agents to a single powerful agent, GPT-5. We label this model 'Conductor w/ all GPT-5'. We present these results in Table 10. We find the Conductor outperforms its "only GPT5" counterpart across the tasks, confirming that our model is indeed harnessing the differing capabilities of the diverse worker pool through its targeted agent selection. In particular, we see that for tasks where GPT-5 already outperforms other closed source models such as AIME, the Conductor with all models matches the performance of its 'only GPT-5' variant, demonstrating that the agentic workflows designed deliver the maximum performance attainable by its best constituent model. In contrast, for tasks in which GPT-5 struggles as GPQA-Diamond or BigCodeBench, the Conductor can leverage the performance of other, more capable agents, thereby allowing for significant performance improvements beyond its 'only GPT-5' variant. Nonetheless, we do note that the performance of the 'only GPT-5' variant is already consistently exceeding that of GPT, highlighting how both subtask design and harnessing collective intelligence are indispensable components to our framework.

Table 10: Conductor performance fixing all agents to GPT-5

| Model | AIME | BigCodeBench | GPQA-D | Avg. |
|---|---|---|---|---|
| Claude Sonnet 4 | 74.30 | 37.16 | 77.70 | 63.0533 |
| Gemini 2.5 Pro | 78.30 | 37.51 | 84.80 | 66.8700 |
| GPT-5 | 90.80 | 32.75 | 82.30 | 68.6167 |
| Conductor w/ all GPT-5 | 93.33 | 33.50 | 82.60 | 69.8100 |
| **Conductor** | **93.30** | **37.86** | **87.50** | **72.8867** |

**Conductor ablation.** We ablate the Conductor itself, replacing our trained Conductor with a powerful frontier model and instructing it to act as a task planner and coordinator, using our identical

Table 11: Replacing our trained Conductor with GPT-5 and Gemini 2.5 Pro.

| Model | LCB | AIME | BigCodeBench | GPQA-D | Avg. |
|---|---|---|---|---|---|
| GPT-5 conduct 7 models | 50.86 | 76.67 | 34.50 | 77.78 | 59.9525 |
| GPT-5 conduct | 67.43 | 93.30 | 33.10 | 86.36 | 70.0475 |
| Gemini 2.5 Pro conduct | 70.29 | 93.30 | 35.13 | 87.62 | 71.5850 |
| **Conductor** | **83.93** | **93.30** | **37.86** | **87.50** | **75.6475** |

prompting setup and overall Conductor framework. We assess three additional models for this ablation. The first, labelled "GPT-5 conduct 7 models", tasks GPT-5 to act as the conductor, allowing it to design coordination strategies of up to 5 workflow steps with access to the same full set of 7 workers. The second and third labelled "GPT-5 conduct 3 models" and "Gemini conduct 3 models", further restrict the agent pool to only the best-performing large frontier models (GPT 5, Gemini 2.5 Pro, Claude Sonnet 4), as we noticed an over-reliance on open-source models in highly suboptimal ways following preliminary experiments with "GPT-5 conduct 7 models" (for example by asking R1-distill-Qwen-32B to act as a final checker before returning the solution to the user, often resulting in solution formatting failures).

To make sure these baselines performed at the best of their ability, we also incorporated an automatic resampling strategy whenever GPT-5 or Gemini produced a format failure (with unlimited retry attempts) and doubled the conductor's original output token limit in order to make better use of their reasoning capabilities.

First, in Table 11 we note that the Conductor expectedly maintains its superior performance across tasks, confirming the effectiveness of our training strategy. In particular, both GPT-5 and Gemini appear over-reliant on an initial pool of models based on their prior biases, which fails to match actual downstream performance. A prime example of this is that the baselines fail to understand that Claude is less capable at LiveCodeBench or GPT-5 is less capable at BigCodeBench, leading to deteriorated performance without an effective feedback mechanism to adjust their knowledge misconceptions, as provided by the Conductor's training phase. Second, we interestingly find that, despite having never been trained on these tasks, both "GPT-5 conduct 3 models" and "Gemini conduct 3 models" outperform their constituent agents (GPT5 and Gemini) across numerous tasks, and their performance is also visibly superior to our much smaller base 7B Qwen model before any training. We believe this provides further evidence validating the Conductor's underlying hypothesis that powerful LLMs are inherently suitable to act as effective meta-orchestrators, highlighting the potential of harnessing future and or larger base models as a simple direction to scale our new framework down the line.

**Few-Shot examples.** We ablate the effect of the few-shot examples provided to the Conductor. In this setting, the Conductor prompt is as specified in Figure 13, but with the few-shot examples removed. This ablation serves to isolate the effect of training a Conductor with its generative distribution conditioned to the orchestration task versus training a Conductor with no prior over workable strategies. We find in Table 9 that few-shot conditioning yields substantial performance gains in the Conductor, with a consistent drop in accuracy across all tasks in the ablation model, denoted *w/o Few-Shot*. This result mirrors prior work in SFT coldstarting (), where conditioning the generative distribution of the model before undergoing reinforcement learning has been widely observed to improve performance.

## B.8    ALTERNATE COORDINATION TOPOLOGY

We evaluate the performance of the Conductor under an alternate coordination topology specification scheme, in which the Conductor specifies, for each agent, which positions in the topology should be made visible. That is, rather than specifying `all` or `[]` for each agent, the Conductor can additionally make visible the output of any agent at a position $p$. Hence, the Conductor could specify for some agent their context to be comprised of the outputs of the agents in the 0, 2, and 3 positions as `[0,2,3]`. This is therefore a generalization of our method, permitting the Conductor more fine-grained control over the composition of each LM agent's context as they attempt their subtask.

We present in Table 9 the results of this alternate co-ordination topology specification scheme, denoted by the model *Fine-grained*. Promisingly, we find that the Conductor is able to learn to use the alternate and more complex scheme effectively, discovering how to design tree and chain topologies with their corresponding subtasks. However, we find that ultimately more complex scheme does not produce significant performance gains, and hence we opt for the simpler binary version presented in the main text of the paper. We leave revisiting the complex, fine-grained control over coordination topology to future work, where we hope that with a larger and more intelligent Conductor beyond 7B, additional performance gains may be unlocked through discovery of powerful topologies.

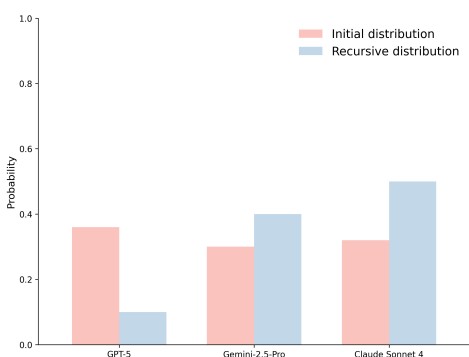

Figure 10: Recursive Conductor worker distribution on BigCodeBench. The Conductor redistributes its agent selection towards Claude and Gemini in recursive rounds, reflecting their superior performance.

### B.9 TASK ADAPTIVITY

We show in Figure 8 that the Conductor is task and difficulty adaptive, where the Conductor learns to allocate more compute to harder tasks and questions. We find that for hard tasks, such as Live-CodeBench code generation problems, the Conductor learns to allocate more compute to solving the problems by devising more worker-intensive coordination strategies, often with multiple planners and then final solvers and verifiers. By comparison, for relatively more straightforward tasks such as information extraction from text or factual recall, the Conductor is more likely to allocate fewer models, or even 1-shot the problem in a single step.

### B.10 RECURSION REDISTRIBUTION

We plot the agent distribution for the Conductor when evaluated on BigCodeBench and permitted to use recursion, as detailed in Section 4.4. We find that the Conductor is able to recognize the suboptimal performance of GPT5 on this task and adaptively redistribute its agent selection towards Claude Sonnet 4 and Gemini 2.5 Pro, illustrating that the Conductor can adapt the coordination strategies learned in pretraining when unseen tasks necessitate it.

## C CONDUCTOR SCHEMATIC

We provide a visualization of our Conductor workflow in Figure 11. In the provided example, we illustrate a task combining translation (between English and Chinese) and mathematical reasoning. The Conductor receives the request in Chinese, calls a Qwen model to first translate the request into English, before calling Gemini to solve the question. Concurrently, the Conductor passes the user question to DeepSeek to solve in Chinese. GPT is called at the end of the workflow to check both the English attempt by Gemini and the Chinese attempt by DeepSeek, before returning the correct solution in Chinese back to the user.

## D RECURSIVE SCHEMATIC

We show in Figure 12 our recursive extension to the Conductor detailed in Section 3.2. With recursion enabled, we allow the Conductor to call itself, providing itself the opportunity to revise the outcome of its previous coordination strategy. In the shown figure, we illustrate an example in which the the Conductor begins with a strategy calling two models, Qwen and Gemini, to first develop a closed form solution to an integral before then applying the bounds. However, upon obtaining the final response of this strategy, the Conductor adapts its strategy to call a new model, DeepSeek, to use numerical integration, obtaining the correct answer to the user query.

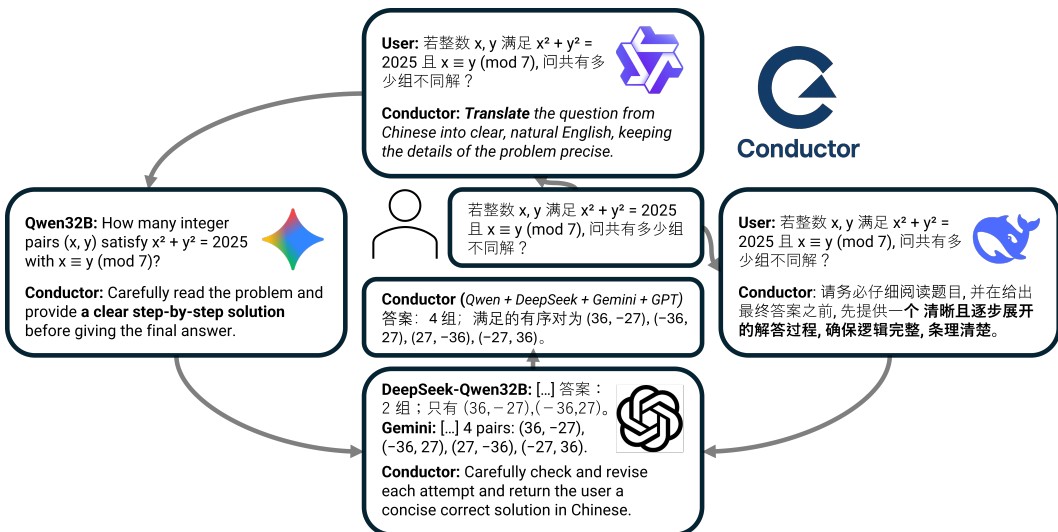

Figure 11: **Conductor schematic visualization**. The Conductor combines the differing specializations of the workers LLMs to answer complex user queries. Here we visualize a workflow bridging both mathematical reasoning and English-Chinese translation.

# E  CONDUCTOR PROMPT AND FEW-SHOT EXAMPLES

We show in Figure 13 our Conductor prompt, which describes in full the Conductor's orchestration task. The core components – consisting of the subtasks, model ids, and access list – are defined to the Conductor, along with the general scheme for how the final response will be obtained from the final model. The available language models are passed to the Conductor as purely ordinal numbers, e.g 'Model 0, Model 1 . . .', in order to fully encourage exploration of the possible models in the pool without possible bias from known model names. We additionally show in Figures 17 and 16 two samples of the few-shot examples we provide to the Conductor to condition its generative distribution to the required orchestration task. These few-shot examples are real Conductor completions taken from coldstart training runs, and are provided as part of the Conductor prompt. Figure 14 shows the recursion prompt, in which we permit the Conductor to devise a new coordination strategy after viewing the outcome of its previous strategy.

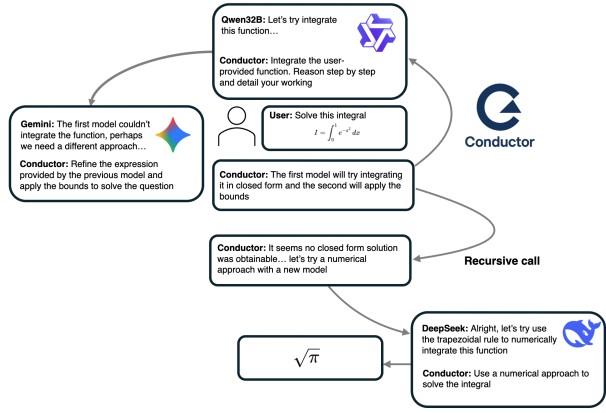

Figure 12: **Recursive Conductor visualization**. At test time, the Conductor is able to adapt its intial coordination strategies on-the-fly.

**Few-shot example details.** As detailed in Section 4, we design our training set as a combination of problems taken from MATH500, LiveCodeBench, MMLU, and RLPR. Our OOD few-shot examples are taken from Medreason (Wu et al., 2025), DeepMath (He et al., 2025), and Countdown (Pan et al., 2025). We select just four examples, one from each of Medreason and DeepMath and two from Countdown, and select examples to ensure a range of workflow steps and selected agents to encourage exploration of the coordination space. Examples shown in Figure 17. For the In-Distribution few-shot setup, we again choose four few-shot examples, with one from each of our component tasks, MATH500, RLPR, MMLU, and LiveCodeBench. Again, we select examples to balance workflow steps and agent selection to encourage exploration. Examples shown in Figure

Figure 13: **The Conductor prompt.** Our Conductor prompt instructs the Conductor with the required format for its output to be parseable as a complete coordination strategy

> *Your role as an assistant involves obtaining answers to questions by an iterative process of querying powerful language models, each with a different skillset.*
>
> *You are given a user-provided question and a list of available numbered language models with their metadata. Your objective is to output a sequence of up to 5 workflow steps.*
>
> *Each routing is made of three elements: A language model, its assigned subtask to accomplish, and an "access list" of past worklow steps it will see in its context when trying to accomplish the subtask.*
>
> *A subtask could directly ask the language model to solve the given question from scratch, refine the solution of the previous subtask in the sequence, or perform any other completely different task that would facilitate later language models in the sequence to answer the original question with their expertise.*
>
> *Based on your answer, the first model selected will be prompted with the user question and the first subtask you define. Each following model in the sequence will be prompted with the history of the previous subtask and response messages specified in its access list, and will be asked to accomplish its relative subtask. The answer of the final model and subtask will be provided back as the final solution to the user.*
>
> *Your response should be provided as three Python lists.*
>
> *The first list should be called model_id, and contain the integers corresponding to the numbered language models in the sequence you want to prompt.*
>
> *The second list should be called subtasks, and contain the strings that will be used to prompt the corresponding language model specified in model_id.*
>
> *The third list should be called access_list, and contain the lists of past routing messages (subtasks and assistant responses) from the previous routing steps to include in the context in the current routing step.*
>
> *You can pass the string all for any of the routing steps in access_list to provide all the previous routing messages in the language model's context. Alternatively, if you want an agent to attempt it's subtask without any access to previous routing steps, you can pass an empty list.*
>
> *For instance:*
>
> *{few-shot examples}*
>
> *USER QUESTION:*
>
> *{user_question}*
>
> *AVAILABLE LANGUAGE MODELS:*
>
> *{available_models}*

16. For the semi-OOD setting, we balance in-distribution and OOD, taking one example from MATH500, one example from LiveCodeBench, one example from Countdown, and one example from Medreason.

## F    EXAMPLE CONDUCTOR COMPLETIONS

We present in this section additional examples of Conductor completions, exemplifying the Conductor's capability as an agentic workflow coordinator. We provide examples including 1) the Conduc-

Figure 14: **The recursive prompt.** When allowing self-referential recursion, the Conductor views the response obtained from its previously designed coordination strategy and decides whether to iterate on its strategy or pass the existing response back to the user.

> *Here is the final response obtained at the end of your routing steps:*
>
> *{worker_response}*
>
> *You now have a chance to correct or improve this response by outputting a new sequence of up to {max_number_of_routing_steps} routing steps, with the same format.*
>
> *Once again, the goal is to produce a final response that answers the original user question correctly. Now, if you pass the string "all" for any of the routing steps in "access_list", the previous final routing message will also be included in the language model's contexts, together with the history of the previous subtask and response messages specified in your new access list.*
>
> *In case the previous final response obtained from your previous routing steps is already correct, you can pass three empty lists for "model_id", "subtasks", and "access_list" to return this to the user as is. In case you think the previous final response is incorrect or in need of verification, you can devise a sequence of routing steps that will revise or verify the previous response.*

tor recognizing the simplicity of a problem and 1-shotting it with a single model (Figure 18), 2) the Conductor organizing the agents into a tree topology to take advantage of independently solvable steps (Figure 19), and 3) the Conductor allocating more compute to a hard problem (Figure 20).

We further illustrate in Figure 2 an example 3B completion which illustrates our claim in Figure 7, that despite the 3B Conductor converging on the same agent selection as the 7B model, its less advanced reasoning capabilities produce less intelligent subtasks.

We additionally present example completions for when the Conductor is permitted to use recursion (Section 3.2), in which we show an example of the Conductor recognizing the response obtained at the end of the initial coordination strategy looks satisfactory and thereby directly returns it (Figure 21) and an example of the Conductor calling an additional agent to check through the past reasoning traces and revise the obtained response (Figure 22).

## F.1 CONDUCTOR CATEGORIZATION

We present in this subsection categorization for some of the most frequent orchestration modes we tended to observe throughout Conductor training and evaluation. We note that these categories are rough, as the collaborative strategies employed by the Conductor are numerous and continue to grow as we evaluate the Conductor in new settings. Nonetheless, we observed frequently the following:

- Sequential coordination with a high-level planner, followed by an executor, usually with additional models in between acting as refiners and checkers (Figures 23, 24)
- Tree coordination, where the first agents work independently and an additional agent aggregates the independent work (Figures 19).
- Tree coordination, where the conductor recognizes the question depends purely on factual recall, and thereby does not necessitate any agent-to-agent collaboration (Figure 27).
- Sequential coordination with a highly logical reasoning process, where each agent proceeds step-by-step to decompose a challenging multi-step problem (Figure 20).
- Conductor task abdication (one of our personal favorites), where the Conductor passes its own role onto one of the more powerful agent models, e.g, asking Gemini 2.5 Pro to come up with subtasks and direct the other models what to do (Figure 28)

Figure 15: **Example 3B Conductor completion.** The 3B model provides a workable strategy, but suboptimally instructs the first model to hide their reasoning due to the constraint provided by the user, impairing collaboration.

**User**

*Solve the following math problem step by step: Let $F(z) = \frac{z+i}{z-i}$ for all complex numbers $z \neq i$, and let $z_n = F(z_{n-1})$ for all positive integers $n$. Given that $z_0 = \frac{1}{137} + i$, find $z_{2002}$. For your final answer, follow these guidelines:*

*1. Provide ONLY the final result without explanations, equations, or steps*
*2. For equations with a single solution, provide just the value (e.g., '5' not 'x=5')*
*...*

**3B Conductor**

*Here's the approach:*

*model_id = [0, 1, 2]*

*subtasks = ["Solve the problem step by step and provide the final value of $z_{2002}$ without explanations, equations, or steps in <idea> tags. Use standard LaTeX notation for the final result.",*

*"Verify the proposed solution by reapplying the function F(z) to check if it indeed leads to the same result after 2002 iterations. Use '<idea> ... </idea>' to show your verification steps and present the final correct answer according to the question's instructions.",*

*"Ensure the result is accurate and provide it in LaTeX notation as instructed."]*

*access_list = [[], ["all"], ["all"]]*

*This sequence divides the work:*
*1. Model 0 directly tackles solving the recurrence relation and providing the final value.*
*2. Model 1 verifies the result by applying the function repeatedly to confirm it converges as expected.*
*3. Model 2 ensures the accuracy of the solution and formats it according to the guidelines.*

Figure 16: **In-distribution few-shot examples.** The two few-shot examples provided are taken from MATH500 and MMLU.

---

*EXAMPLE 1:*

*Question: Subject: Physics. A converging lens is placed 30.0 cm to the right of a diverging lens of focal length 10.0 cm. A beam of parallel light enters the diverging lens from the left, and the beam is again parallel when it emerges from the converging lens. Calculate the focal length of the converging lens.*

*Assistant Response: Given the simplicity of the problem and the required formatting, we'll use two models that can handle physics calculations. Model 3 can perform the required physics calculation and determine the focal length of the converging lens based on the given conditions.*
*" "Subsequently, Model 5 can verify the solution, format the result according to the required guidelines, and return the final response.*

*model_id = [3, 5]*

*subtasks = ["Calculate the focal length of the converging lens and provide the final result in the required format. Show your working in <idea> </idea< tags. ",*

*"Verify the solution proposed by the previous model and refine if necessary. Format the final answer in the required format, using standard mathematical notation and units."]*

*access_list = [[], ["all"]]*

*EXAMPLE 2:*

*Question: Solve the following math problem step by step: A palindrome is a number that reads the same forwards and backwards. The sum of a particular set of three consecutive positive integers is a three-digit palindrome. If the sum is less than 220, what is the greatest possible value for the largest of the three integers in the set?.*

*Assistant Response: We'll need at least 3 models to solve this. We can use model 0 to first try and solve the question, model 2 can check if the answer is correct given the constraints, and then with the additional context, model 0 can finalize and re-attempt the problem if necessary.*

*model_id = [0, 2, 0]*

*subtasks = ["Find a set of three consecutive positive integers whose sum is a three-digit palindrome smaller than 220 and calculate the largest of those numbers.",*

*"Verify if the number found by the previous agent is the largest set of three consecutive positive integers whose sum is a three-digit palindrome smaller than 220.",*

*"Ensure the final answer is formatted correctly according to the rules and provide the necessary LaTeX representation."]*

*access_list = [[], ["all"], ["all"]]*

---

Figure 17: **Out-of-distribution few-shot examples.** The two few-shot examples provided are taken from deepmath and medreason, which are out-of-distribution relative to the training tasks of MATH500, MMLU, RLPR, and LiveCodeBench.

*EXAMPLE 1:*

*Question: Does brain-derived neurotrophic factor enhance the contraction of intestinal muscle strips induced by SP and CGRP in mice? Answer Choices: A. Yes B. No*

*Assistant Response: The user question is about determining if brain-derived neurotrophic factor (BDNF) enhances the contraction of intestinal muscle strips induced by SP and CGRP in mice. This seems like a straightforward query that may not require iterative modeling as it's more about the availability of scientific data or a standard result from experimentation rather than problem solving or algorithmic implementation. The best approach will be to directly ask each model to provide an answer, as the user question is likely based on known data rather than needing complex model-to-model collaboration.*

*model_id = [1, 0, 2]*

*subtasks = ["Does brain-derived neurotrophic factor enhance the contraction of intestinal muscle strips induced by SP and CGRP in mice? Answer with A for Yes or B for No.",*

*"Does brain-derived neurotrophic factor enhance the contraction of intestinal muscle strips induced by SP and CGRP in mice? Answer with A for Yes or B for No",*

*"Check the two previous answers and provide the correct answer according to the question's formatting instructions if necessary."]*

*access_list = [[], [], ["all"]]*

*EXAMPLE 2:*

*Question: Evaluate the limit:*

$$\lim_{t \to 0} \left( \frac{1}{\ln(1+t)} + \frac{1}{\ln(1-t)} \right).$$

*Provide the final answer in <answer> </answer> tags and use LaTeX notation.*

*Assistant Response: Given the difficulty of the question, let's try 4 models to solve this. The first two models can work independently to propose a limit as t tends to 0 by potentially using a Taylor expansion, the third model can check the work of the first two models and optionally use L'Hopital's rule to confirm the result, and the final model can oversee the entire process, handle final verification by confirming that all approaches point to the same conclusion and return the response to the user.*

*model_id = [6, 5, 3, 2]*

*subtasks = ["Understand the question and provide an intial solution to approximate the limit as t tends to 0, potentially by using a Taylor expansion. Show your work in <idea> </idea> tags.",*

*"Understand the question and provide an intial solution to approximate the limit as t tends to 0, potentially by using a Taylor expansion. Show your work in <idea> </idea> tags.",*

*"Verify the work done by the first two models and optionally use L'Hopital's rule or numerical methods to confirm the result. Show your work in <idea> </idea> tags.",*

*"Check the work of the previous models, refine where necessary and obtain the correct final answer. Provide the final answer according to the question's formatting instructions."]*

*access_list = [[], [], ["all"], ["all"]]*

Figure 18: **Example Conductor completion with a 1-shot strategy.** In the following example, the Conductor recognizes the simplicity of the question and directly solves it with a single model.

**User**

*A heartbroken dog whose owner died two months ago is missing her so much that he attends services every day at the Italian church where her funeral was held, patiently waiting for her to return. Tommy, a seven-year-old dog, belonged to Maria Lochi, 57, and had been her faithful partner after she adopted him when she found him in fields close to her home.*

*...*

*The dog Tommy still goes to the church every day to ˍ.*

*Options:*
*A. attend his owner's funeral*
*B. wait for Lochi to come back*
*C. stay together with Lochi*
*D. celebrate Mass with Father Panna*

**Conductor**

*This is a straightforward comprehension and recall task from the given text, so we only need one model to do that: 1. First, comprehend and recall the information from the story provided for the dog Tommy, which requires an understanding of the context and deducing Tommy's reason for going to the church. 2. Output the most accurate answer from the comprehension of the context.*

*model_id = [3]*
*subtasks = ["Deduce from the story provided in the user question, why Tommy still goes to the church every day to the most appropriate option among the given choices A, B, C, and D. Provide the option letter in <answer> tags, for example <answer>B</answer>."]*
*access_list = [[]]*

Figure 19: **Example Conductor completion with a tree topology.** The Conductor solves the problem with two independent steps followed by an aggregation step, specified by its access list of `[[], [], ["all"]]`.

**User**

*One ticket to a show costs* $20 *at full price. Susan buys* 4 *tickets using a coupon that gives her a* 25% *discount. Pam buys* 5 *tickets using a coupon that gives her a* 30% *discount. How many more dollars does Pam pay than Susan?*

**Conductor**

*1. We need to calculate how much Susan pays for her 4 discounted tickets (Model 1). 2. We need to calculate how much Pam pays for her 5 discounted tickets (Model 2). 3. We need to determine the difference in cost between Pam's and Susan's tickets (Model 3).*

*model_id = [1, 2, 3]*
*subtasks = ["Calculate the cost of 4 tickets at a 25% discount.",*

*"Calculate the cost of 5 tickets at a 30% discount.",*

*"Determine the difference in cost between what Pam and Susan paid for their tickets."]*
*access_list = [[], [], ["all"]]*

Figure 20: **Example Conductor completion allocating more agents to solve a harder problem.** In the following LiveCodeBench example, the Conductor devises an extensive workflow utilizing all 5 possible workflow steps.

**User**

*You are given a weighted simple connected undirected graph with N vertices and M edges, where vertices are numbered 1 to N, and edges are numbered 1 to M. Additionally, a positive integer K is given. Edge i $(1 \leq i \leq M)$ connects vertices $u_i$ and $v_i$ and has a weight of $w_i$. For a spanning tree T of this graph, the cost of T is defined as the sum, modulo K, of the weights of the edges in T. Find the minimum cost of a spanning tree of this graph.*

*The input is given from Standard Input in the following format:*

$$\begin{bmatrix} u_1 & v_1 & w_1 \\ u_2 & v_2 & w_2 \\ \vdots & \vdots & \vdots \\ u_M & v_M & w_M \end{bmatrix}$$

*Print the answer*

**Conductor**

*1. The first model (Model 6) should understand the problem statement and define the objectives. 2. The second model (Model 4) should analyze the given graph and its edges to understand the structure. 3. The third model (Model 5) should determine the strategy to find the minimum cost spanning tree. 4. The fourth model (Model 3) will implement the strategy in a Python function. 5. The final model (Model 2) will validate the implemented solution, ensuring it meets all constraints and requirements.*

*model_id = [6, 4, 5, 3, 2]*

*subtasks = ["Understand the problem statement and goals. Define what needs to be done to solve the given problem.",*

*"Analyze the graph structure and the constraints to identify key steps in the algorithm for solving the given problem.",*

*"Develop a strategy to find the minimum cost of a spanning tree with costs computed modulo K. Ensure the strategy can handle the input constraints.",*

*"Implement the strategy in a Python function to solve the problem as defined, ensuring all constraints are met.",*

*"Validate and test the Python implementation to ensure the solution is accurate and meets all requirements."]*

*access_list = [[], ["all"], ["all"], ["all"], ["all"]]*

Figure 21: **Example Conductor-recursive completion, determining that the original coordination strategy was already sound.**

**User**

*You are given a string S of length N consisting of lowercase English letters.*

*If there are any adjacent occurrences of a and b in S, print Yes; otherwise, print No. (The order of a and b does not matter.)*

**Conductor**

*1. First, we'll directly attempt to solve the problem. 2. Next, we'll check and refine the solution if necessary. 3. Finally, we'll provide the most accurate and detailed solution, following all the given instructions.*

*model_id = [0, 1, 2]*

*subtasks = ["Transform the string s into t in exactly k operations and provide the number of ways to do so. Show your work in <idea> tags and present the final solution in ¡answer¿ tags.",*

*"Check the proposed solution and refine it if necessary, ensuring all steps are correct and complete. Provide a detailed verification in ¡idea¿ tags and a final solution in <answer> tags.",*

*"Refine the solution further and provide a detailed, accurate, and well-structured solution. Provide the solution in <idea> and ¡answer¿ tags as specified, and implement it in a Python function named 'numberOfWays'."]*

*access_list = [[], ["all"], ["all"]]*

*RESULT OF COORDINATION STRATEGY:*

*Using the plan developed by the previous agents, we can implement the function as follows:*

*from typing import List*

*class Solution:*
    *def numberOfWays(self, s: str, t: str, k: int) -> int:*
*...*

**Conductor-recursive**

*The previous response provided a correct and efficient solution. It seems well-reasoned and directly addresses the user's question with the correct Python code. Given that the model's response is correct and concise, I don't need to make any further attempts or verify it. Therefore, I will pass three empty lists to return the previous correct response to the user.*

*model_id = []*
*subtasks = []*
*access_list = []*

Figure 22: **Example Conductor-recursive completion, determining allocating additional agents to provide feedback and verify in the recursion round.**

**User**

*You are given a string S of length N consisting of lowercase English letters.*

*If there are any adjacent occurrences of a and b in S, print Yes; otherwise, print No. (The order of a and b does not matter.)*

**Conductor**

*1. First, we'll directly attempt to solve the problem. 2. Next, we'll check and refine the solution if necessary. 3. Finally, we'll provide the most accurate and detailed solution, following all the given instructions.*

*model_id = [0, 1, 2]*

*subtasks = ["Take the string and devise an algorithm for counting adjacent occurrences of a given letter. Show your work in <idea> tags and present the final solution in <answer> tags.",*

*"Check the proposed solution and refine it if necessary, ensuring all steps are correct and complete. Provide a detailed verification in <idea> tags and then implement the algorithm when applied to 'a' and 'b' in Python in <answer> tags.",*

*"Refine the solution further and provide a detailed, accurate, and well-structured solution. Provide the solution in <idea> and <answer> tags as specified, and implement it in a Python function named 'numberOfWays'."]*

*access_list = [[], ["all"], ["all"]]*

*RESULT OF COORDINATION STRATEGY:*

*Using the plan developed by the previous agents, we can implement the function as follows:*

*from typing import List*

*class Solution:*
    *def numberOfWays(self, s: str, t: str, k: int) -> int:*
*...*

**Conductor-recursive**

*The previous response seems to be correct, as it handles the presence of "ab" or "ba" in the stripped input string 's'. However, to ensure robustness, I'll ask one model to provide feedback and another to verify the solution.*

*model_id = [0, 1]*

*subtasks = ["Verify the previous solution and provide your feedback in <idea> tags. If correct, directly return the solution. If incorrect, correct it and return the final code in <answer> tags.",*

*"Review the previous model's work, verify its solution, and provide the final correct answer according to the user requirements. Show your reasoning in ¡idea¿ tags and return the final code in <answer> tags."]*

*access_list = [["all"], ["all"]]*

Figure 23: **Utilizing a planner and coder in LiveCodeBench.** In this LiveCodeBench problem, the Conductor leverages Gemini 2.5 Pro to first act as a strategist and then GPT to act as a coder.

**User**

*There are N cities in a certain country.*
*You will travel from your office in city 1 to a destination in city N, via zero or more cities.*
*Two types of transportation are available: company car and train. The time required to travel from city i to city j is as follows:*
*. . .*

**Conductor**

*First, we need to understand the problem and the transportation options available. Then we need to find the minimum time to travel from city 1 to N, considering the various options. Finally we need to implement the solution in Python and ensure it meets all the constraints and requirements.*
*model_id = [0, 2]*

*subtasks = ["Understand the problem and the transportation options available. Determine the strategy to find the minimum time to travel from city 1 to city N",*

*"Implement the solution in Python according to the problem constraints and requirements and provide the final code in <answer>tags. Show your work in <idea> tags if needed."]*

*access_list = [[],[["all"]]]*

Figure 24: **Comprehending, planning, strategizing, and finally coding with 3 agents on Live-CodeBench.** In this LiveCodeBench problem, the Conductor leverages Claude Sonnet 4 to first understand the objective and constraints, then Gemini 2.5 Pro strategizes and plans, before GPT-5 implements the final code.

**User**

*You are given a 0-indexed integer array nums representing the initial positions of some marbles. You are also given two 0-indexed integer arrays moveFrom and moveTo of equal length.*

*...*

*After completing all the steps, return the sorted list of occupied positions.*

**Conductor**

*First, we need to understand the problem, its constraints, and the goal. Then, we need to come up with an efficient strategy for how to solve. Finally we need to write optimized code implementing the plan.*

*model_id = [1, 0, 2]*

*subtasks = ["Understand the problem and determine the goal, which is to implement a function that returns the total frequencies of elements with the maximum frequency in a given array. Also, analyze the given examples and constraints to understand the best approach.",*

*"Analyze the examples and constraints to find a strategy for determining the elements with the maximum frequency and calculating their total frequency. Provide a step-by-step plan in <idea> tags."*

*"Implement the Python function 'maxFrequencyElements' following the strategy from the previous step. Show your work in <idea> tags and return the final code in <answer> tags as specified."]*

*access_list = [[],["all"],[["all"]]*

Figure 25: **Understanding the method, analyzing the data, then solving in MMLU.** In this MMLU problem, the Conductor leverages Gemini 2.5 Pro and Claude Sonnet 4 to analyze the data and methods, before handing over to GPT-5 to aggregate their work into the final response.

**User**

*Flying over a desert in an airplane, two scientists looked down with trained eyes at trees and bushes. After an hour's flight one of the scientists wrote in his notebook, "Look here for probable metal."...*
*...*
*The study of trees, branches and roots indicated that _ .*
*...*

**Conductor**

*First, we need to understand the context and scientific method used by the scientists to locate minerals based on the condition of plants and trees. At the same time, we need to analyze specific data provided about gold in different parts of the trees to determine the correct answer.*
*model_id = [0, 1, 2]*

*subtasks = ["Understand the scientific method used by the scientists to locate minerals based on tree and plant characteristics.",*

*"Analyze the data collected from the trees, branches, and seeds to determine the gold distribution"*

*''Based on the methodology and the data, analyze which option is correct and provide the solution in <answer> tags"]*

*access_list = [[],[],[["all"]]*

Figure 26: **Factual recall makes use of tree topology.** In this Medreason problem, the Conductor explicitly recognizes that agent-to-agent collaboration is unnecessary, and relies instead on querying models independently and aggregating to solve the answer.

**User**

*Does brain derived neurotrophic factor enhance the contraction of intestinal muscle strips induced by SP and CGRP in mice?*

**Conductor**

*The user question is about determining if brain-derived neurotrophic factor (BDNF) enhances the contraction of intestinal muscles stripes induced by SP and CGRP in mice. This seems like a straightforward query that may not require iterative modeling as its more about the availability of scientific data or a standard result from experimentation rather than problem solving or algorithmic implementation.*
*The approach will be to directly ask two models to provide an answer, as the user question is likely based on known data rather than needing complex model-to-model collaboration.*
*model_id = [0, 2, 1]*

*subtasks = ["Does brain derived neurotrophic factor enhance the contraction of intestinal muscle strips induced by SP and CGRP in mice?",*

*"Does brain derived neurotrophic factor enhance the contraction of intestinal muscle strips induced by SP and CGRP in mice?"*

*''Check the two previous answers to the user question and provide the correct answer according to the formatting instructions if necessary."]*

*access_list = [[],[],[["all"]]]*

Figure 27: **Small models can help format check.** In this BigCodeBench problem, we noticed GPT-5 making frequent formatting errors, providing additional information and hyerparameter choices in the dataframe title. Employing Qwen-32B as a final format checker helped solve this problem and raise GPT-5's performance.

**User**

*Create a Pandas DataFrame from a list of pairs and visualize the data using a bar chart. - The title of the barplot should be set to 'Category vs Value''.*
*The function should output with:*
*..*

**Conductor**

*We'll have 1 model plan out the requirements for the visualization and how to do the sampling, and a second to implement the code as required. We'll use a final model to ensure the answer abides by the user's formatting requirements.*
*model_id = [0, 2, 3]*

*subtasks = ["Consider the user question and plan out the key aspects of the visualization. Use <idea> tags to show your working and ideas.",*

*"Create the required Pandas DataFrame using the ideas provided by the first model. Provide your answer in <answer> tags."*

*"Ensure the final code satisfies the exact user-provided formatting requirements and constraints before returning the code back to the user."]*

*access_list = [[],["all"],[["all"]]*

Figure 28: **Conductor role abdication.** The Conductor abdicates its role as a subtask planner to Gemini 2.5 Pro, tasking it with devising subtasks for the other models to enact.

**User**

*You are given an array nums consisting of positive integers. We call a subarray of an array complete if the following condition is satisfied:*

*The number of distinct elements in the subarray is equal to the number of distinct elements in the whole array...*
*...*
*Implement the function 'countCompleteSubarrays' that solves this problem.*

**Conductor**

*We'll need models with strategic and analytical skills to implement an efficient algorithm, validate our logic, and handle Python code. Let's leverage Model 0 (Analytical) to dissect the problem and plan the subtasks. The other models will then follow these subtasks to solve the problem.*

*Based on this, I have a sequence of three models that each will contribute to solving the problem but especially filling in the algorithm and Python code.*
*model_id = [0, 1, 2]*

*subtasks = ["Analyze the problem, understand the constraints, and propose a strategy to solve the problem. Break down the subtasks step by step, including identifying necessary variables and the logic involved.",*

*"Follow the plan and assigned subtask provided by the first model"*

*"Based on the assigned subtask provided by the first model, identify any runtime errors or issues with the algorithm and return the final correct Python function to the user."]*

*access_list = [[],["all"],[["all"]]]*

