# OpenReview forum: "Learning to Orchestrate Agents in Natural Language with the Conductor"
_ICLR.cc/2026/Conference — ICLR 2026 Poster_

### Official Review · Reviewer_og9K · 2025-10-19

**Soundness:** 3
**Presentation:** 3
**Contribution:** 3
**Rating:** 8
**Confidence:** 2

**Summary:**

This paper introduces the conductor, which can automatically divide the task and assign subtasks to different models, and a method to training the conductor with RL. With the conductor, small LLMs surpass large LLMs on multiple tasks.

**Strengths:**

- The concept of conductor is novel. The workflow is fully automatic and doesn't need human's design.
- The quantitive result is promising.
- The extra analysis is detailed.

**Weaknesses:**

N/A

**Questions:**

N/A

---

> ### Author Response · Authors · 2025-11-20
> **Response to reviewer (1/2)**
>
> We greatly appreciate the reviewer’s endorsement of our work. In case the reviewer is curious, we’ve provided below some additional empirical results addressing the performance-efficiency benefits of our framework, and additional ablations which help demonstrate the Conductor’s ability to learn the true underlying capabilities of the agents and how to combine them in powerful coordination strategies.
>
> If the reviewer has any further questions or comments, we hope they will not hesitate to let us know.
>
> Table A1
>
> | Model                          | Performance | Token Usage | Cost      | Cost-adjusted performance |
> |--------------------------------|-------------|-------------|-----------|---------------------------|
> | Claude Sonnet 4 5x consensus   | 91          | 1412.8      | 0.021192  | 42.94073235               |
> | Claude Sonnet 4 5x reflect     | 90.66       | 2516.992    | 0.02080128| 43.58385638               |
> | Gemini 2.5 Pro 5x consensus    | 91.6        | 1658.4      | 0.016584  | 55.23396044               |
> | Gemini 2.5 Pro 5x reflect      | 88.33       | 2919.776    | 0.01675976| 52.70361867               |
> | GPT 5 5x consensus             | 91.3        | 1376.3      | 0.013763  | 66.33728112               |
> | GPT 5 5x reflect               | 91.79       | 2457.132    | 0.01424907| 64.41823923               |
> | Conductor                      | 93.14       | 735.2       | 0.009     | 103.4888889               |
>
>
>
> Table A2
>
> | Model   | Performance | Token Usage | Cost |
> |----------------|---------|----------|------------------|
> | MoA            | 62.13 | 11203 |  $0.048554  |
> | Smoothie    | 56.48  |9909   | $0.039291  |
> | RDC            | 52.41  | 840    | $0.005613 |
> | MasRouter  | 56.89 | 4970   | $0.01345   |
> | Conductor   | 72.35 | 1820   | $0.02384   |
>
>
> We present above further efficiency results in Tables A1 and A2, where we show the performance, token usage, and average cost of the Conductor in comparison to the multi-agent baselines (Smoothie, RouterDC, MoA, MASRouter. We additionally present results comparing the Conductor with single-agent inference-time scaling frameworks, consensus [1], and self-reflection. We see, as with Figure 5, that the substantially lower API calls of the Conductor similarly translate into lower token consumption, lower cost, and further improved cost-adjusted performance, demonstrating the gains in both performance and efficiency of our approach.

---

> ### Author Response · Authors · 2025-11-20
> **Response to reviewer (2/2)**
>
> Table B
>
> | Model                     | LCB   | AIME | BigCodeBench | GPQA-D | Avg.    |
> |---------------------------|-------|------|--------------|--------|---------|
> | GPT-5 conduct 7 models   | 50.86 | 76.67 | 34.5       | 77.78 | 59.9525  |
> | GPT-5 conduct  3 models           | 67.43 | 93.3 | 33.1         | 86.36  | 70.0475 |
> | Gemini 2.5 Pro conduct  3 models  | 70.29 | 93.3 | 35.13        | 87.62  | 71.585  |
> | Conductor                 | 83.93 | 93.3 | 37.86        | 87.5   | 75.6475 |
>
> Table C
>
> | Model                  | AIME | BigCodeBench | GPQA-D | Avg.        |
> |------------------------|------|--------------|--------|-------------|
> | Claude Sonnet 4        | 74.3 | 37.16        | 77.7   | 63.05333333 |
> | Gemini 2.5 Pro         | 78.3 | 37.51        | 84.8   | 66.87       |
> | GPT-5                  | 90.8 | 32.75        | 82.3   | 68.61666667 |
> | Conductor with only GPT-5 | 93.33| 33.5         | 82.6   | 69.81       |
> | Conductor              | 93.3 | 37.86        | 87.5   | 72.88666667 |
>
> First, in Table C, we see that the Conductor outperforms this additional model across the tasks, demonstrating that the Conductor is indeed harnessing the differing capabilities of the diverse worker pool through its agent selection. Second, we see that for tasks in which GPT-5 is already dominant (for example, AIME), the Conductor matches the performance of the ‘only GPT-5’ model, demonstrating that the agentic workflows designed by the Conductor deliver the maximum performance attainable by its best constituent model. Crucially, we also see that for tasks in which GPT-5 struggles (for example, in GPQA-Diamond or BigCodeBench), the Conductor is able to leverage the performance of other more capable agents, thereby allowing the Conductor to deliver performance substantially beyond the ‘only GPT-5’ model. This therefore demonstrates that the Conductor, through its agent selection, is indeed aggregating the differing advantages and capabilities across its agent pool. Finally, we note as well that the performance gain between GPT-5 and Conductor with only GPT-5 is small. This highlights another important point, which is that for powerful frontier models at maximum reasoning and output budgets, the task of designing agentic workflows to deliver additional performance gains across a range of tasks is highly non-trivial, and indeed, this echoes much of the findings regarding self-reflection-based scaffolds. We therefore point to the performance gains accessed by the Conductor as strong evidence of the intelligent collaborative modes and optimal agent selection discovered by our setup.
>
> In Table B, we note that the Conductor expectedly maintains its superior performance across tasks, confirming the effectiveness of our training strategy. In particular, both GPT-5 and Gemini appear over-reliant on an initial pool of models based on their prior biases, which fails to match actual downstream performance. A prime example of this is that the baselines fail to understand that Claude is less capable at LiveCodeBench or GPT-5 is less capable at BigCodeBench, leading to deteriorated performance without an effective feedback mechanism to adjust their knowledge misconceptions, as provided by the Conductor's training phase. Second, we interestingly find that, despite having never been trained on these tasks, both "GPT-5 conduct 3 models" and "Gemini conduct 3 models" outperform their constituent agents (GPT5 and Gemini) across numerous tasks, and their performance is also visibly superior to our much smaller base 7B Qwen model before any training. We believe this provides further evidence validating the Conductor's underlying hypothesis that powerful LLMs are inherently suitable to act as effective meta-orchestrators, highlighting the potential of harnessing future and or larger base models as a simple direction to scale our new framework down the line.
>
> [1] Wang, Xuezhi, et al. "Self-consistency improves chain of thought reasoning in language models." International Conference on Learning Representations (ICLR) (2023)

---

### Official Review · Reviewer_c7Bs · 2025-10-31

**Soundness:** 3
**Presentation:** 4
**Contribution:** 2
**Rating:** 4
**Confidence:** 4

**Summary:**

The paper presents a major conceptual step: turning a small LLM into a meta-agent that learns to orchestrate larger, specialized LLMs through reinforcement learning. Instead of directly solving problems, the Conductor learns to design agentic workflows—decomposing tasks into subtasks, assigning them to specialized worker models, and defining how agents communicate and share context.
It establishes new SOTA reasoning results on tasks such as LiveCodeBench, GPQA-Diamond, and AIME 2025 and introduces a flexible, extensible framework for autonomous multi-agent coordination via language.

**Strengths:**

- While prior work (e.g., Mixture-of-Agents [Wang et al., 2024], or Smoothie [Guha et al., 2024]) explores routing or fixed multi-agent topologies, this work chooses to learn such coordination end-to-end using pure reward maximization, without predefined roles or handcrafted scaffolds.
- The idea of recursively calling itself as part of the agentic workflow is interesting and open pathways for further studies.
- The empirical results are strong, beating top frontier models.
- I like the in-domain and out of domain analysis. Really put things in perspectives. It is nice to know that Conductor is not just overfitting on specific task or formats.

**Weaknesses:**

```One concern I have is whether the benefits from Conductor came from orchestration or agentic workflow planning, or purely prompt engineering?```

The Conductor orchestrates the agentic flow with respective subtasks. And RL is used to train the model to do better at this task which can be roughly divided to two tasks: 1) manage and assign the best match between model and tasks. 2) create better subtask directions/prompts. There is no ablation that fixes 1), that is, use only one model (like GPT-5) and then train Conductor with RL. If Conductor is still doing pretty well, that means benefits from orchestrating is minimal. I would highly suggest doing such an ablation which would make your case a lot stronger.

```The tasks themselves are not very multi-agent dependent```

Tasks like Math500, AIME or LiveCodeBench are not really suitable tasks for multi-agent. Ideally, a multi-agent task or a task where multi-agent can have some leverage or just single agent are tasks that naturally can be divided to subtasks that are heterogenous. I don't find these tasks to have those properties. Like do you write first part of the math solution and then write the second part using another model? I saw some examples where Conductor ask models to refine or verify. But those are not really something that a model have specialty in. The task selection is a bit off.

```Lack of other benefits```

To continue from last point. A weird task selection would cause the benefit of multi-agent to diminish. Multi-agent frameworks comes in with three big pros -- efficiency, safety and performance. There is no efficiency benefits for this because everything is sequential and one model has to wait for last model to finish (correct me if I misunderstood). This limits the contribution of the Conductor. I would love to see such a pipeline work in a meaningful agentic setup.

**Questions:**

Typos:
- Line 200 "to its own parent output"

Question:

- How sensitive is the Conductor’s learning to the choice of reward granularity (binary correctness vs. partial credit or verification-based rewards)? Would denser reward shaping improve stability or lead to different emergent strategies? What reward strategies have you guys tried? I am generally really curious about this.

- What are the different modes of orchestration the Conductor learned? Did you guys do some simple categorization?

---

> ### Author Response · Authors · 2025-11-20
> **Response to reviewer (1/6)**
>
> > **Weakness 1**: One concern I have is whether the benefits from Conductor came from orchestration or agentic workflow planning, or purely prompt engineering?** The Conductor orchestrates the agentic flow with respective subtasks. And RL is used to train the model to do better at this task which can be roughly divided to two tasks: 1) manage and assign the best match between model and tasks. 2) create better subtask directions/prompts. There is no ablation that fixes 1), that is, use only one model (like GPT-5) and then train Conductor with RL. If Conductor is still doing pretty well, that means benefits from orchestrating is minimal. I would highly suggest doing such an ablation which would make your case a lot stronger.
>
> We thank the reviewer for suggesting this valuable additional baseline, which will help isolate the benefit of the Conductor’s learned agent selection. Thus, following their suggestion, **we added new results for an additional model, labelled ‘Conductor w/ only GPT-5’, in which we instantiate the Conductor's framework with only GPT-5 while allowing the Conductor to still design workflows of up to 5 steps, with full freedom to specify the subtask assignment and access list as before**. We present evaluations across BigCodeBench, GPQA-Diamond, and AIME25 and a summary of our analysis below.
>
> As reported in Table C below, the Conductor outperforms its "only GPT-5" counterpart across the tasks, confirming that our model is indeed harnessing the differing capabilities of the diverse worker pool through its targeted agent selection. In particular, we see that for tasks where GPT-5 already outperforms other closed-source models, such as AIME, the Conductor with all models matches the performance of its ‘only GPT-5’ variant, demonstrating that the agentic workflows designed deliver the maximum performance attainable by its best constituent model. In contrast, for tasks in which GPT-5 struggles as GPQA-Diamond or BigCodeBench, the Conductor can leverage the performance of other, more capable agents, thereby allowing for significant performance improvements beyond its ‘only GPT-5’ variant. Nonetheless, we do note that the performance of the ‘only GPT-5’ variant is already consistently exceeding that of GPT, highlighting how both subtask design and harnessing collective intelligence are indispensable components to our framework.
>
> Table C
> | Model                  | AIME | BigCodeBench | GPQA-D | Avg.        |
> |------------------------|------|--------------|--------|-------------|
> | Claude Sonnet 4        | 74.3 | 37.16        | 77.7   | 63.05333333 |
> | Gemini 2.5 Pro         | 78.3 | 37.51        | 84.8   | 66.87       |
> | GPT-5                  | 90.8 | 32.75        | 82.3   | 68.61666667 |
> | Conductor w/ only GPT-5 | 93.3| 33.50         | 82.6   | 69.81       |
> | Conductor              | 93.3 | 37.86        | 87.5   | 72.88666667 |
>
> **We’ve added these additional results to our updated manuscript in Appendix B** which we hope will provide a valuable extension to our analysis and make the Conductor’s motivation stronger.

---

> ### Author Response · Authors · 2025-11-20
> **Response to reviewer (2/6)**
>
> > **Weakness 2** : The tasks themselves are not very multi-agent dependent** Tasks like Math500, AIME or LiveCodeBench are not really suitable tasks for multi-agent. Ideally, a multi-agent task or a task where multi-agent can have some leverage or just single agent are tasks that naturally can be divided to subtasks that are heterogenous. I don't find these tasks to have those properties. Like do you write first part of the math solution and then write the second part using another model? I saw some examples where Conductor ask models to refine or verify. But those are not really something that a model have specialty in. The task selection is a bit off.
>
> We break the reviewer’s concern into two parts:
>
> > **a**: The task selection is a bit off.
>
> We note that many of our tasks are used in other popular multi-agent works. For example, Math500 is used in [1] and MMLU is used in [1,2]. We also note that we chose harder versions of popular benchmarks in multi-agent papers. For example, [1,2,3] use GSM8K to evaluate mathematical reasoning, and we chose the more challenging AIME25. Overall, we aimed to select a range of tasks not only for their usage in other multi-agent works, but also for their difficulty and popularity in measuring frontier model performance. **Following the reviewer’s concern, we’ve added clarification and extended discussion to Appendix A** which we hope will help clarify the motivation of our task selection.
>
> > **b**: Ideally, a multi-agent task or a task where multi-agent can have some leverage or just single agent are tasks that naturally can be divided to subtasks that are heterogenous. I don't find these tasks to have those properties. Like do you write first part of the math solution and then write the second part using another model?
>
>
> Regarding the reviewer’s point that an ideal multi-agent task is a task that naturally can be divided into heterogeneous subtasks - we absolutely agree, and indeed note that this is actually what we observe for several of the challenging reasoning tasks considered. To help clarify this point, **we’ve added additional examples to Appendix F (Figures 23, 24, 25), which demonstrate the heterogeneous, complementary subtasks performed by the agent models, along with additional discussion of how these challenging tasks can be decomposed into meaningful subtasks.**
>
> For instance, a simple example of this is LiveCodeBench, where strong performance often requires language understanding, mathematical and computational reasoning, algorithm planning and design, and code optimization. Following this intuition, we do find these are the kinds of emergent behaviors we observe the Conductor learning as optimal subtasks, where in many cases one model will be tasked with first understanding the problem and its mathematical properties before then working on an algorithmic sketch of the solution, while a second model will be tasked with taking that algorithm and writing it into optimized code. Examples of this exact behavior can be seen in Figures 2 and 3, which are real Conductor completions. To further support this point, we’ve added three additional example completions into Appendix F (Figures 23, 24, 25), which demonstrate the heterogeneous, complementary subtasks issued by the Conductor that leverage individual worker skillsets.
>
> Nonetheless, if the reviewer has any recommendations for other challenging reasoning tasks that could be better suited to multi-agent setups, we hope they will not hesitate to let us know, and we will start working to implement those as soon as possible.
>
>
> [1] Yanwei Yue, Guibin Zhang, Boyang Liu, Guancheng Wan, Kun Wang, Dawei Cheng, and Yiyan Qi. Masrouter: Learning to route llms for multi-agent systems. ACL (2025)
>
> [2] Shuhao Chen, Weisen Jiang, Baijiong Lin, James Kwok, and Yu Zhang. Routerdc: Query-based router by dual contrastive learning for assembling large language models. Advances in Neural Information Processing Systems, 37:66305–66328, 2024
>
> [3] Yilun Du, Shuang Li, Antonio Torralba, Joshua B Tenenbaum, and Igor Mordatch. Improving factuality and reasoning in language models through multiagent debate. In Forty-first International Conference on Machine Learning, 2023

---

> ### Author Response · Authors · 2025-11-20
> **Response to reviewer (3/6)**
>
> > **Weakness 3**: Lack of other benefits.** To continue from last point. A weird task selection would cause the benefit of multi-agent to diminish.
>
> We break the reviewer’s concern into two parts, addressing first the overall efficiency of the Conductor (a), followed by the point regarding sequential execution (b).
>
> > **a**: Multi-agent frameworks comes in with three big pros -- efficiency, safety and performance. There is no efficiency benefits for this...
>
> To address the reviewer’s concern regarding efficiency, **we conducted additional evaluation of our Conductor alongside both the frontier models when evaluated with consensus inference-time scaling [1] and self-reflection, and the multi-agent baselines in terms of token usage, cost, and cost-adjusted performance**. We present the results in Tables A1 and A2 below, where we indeed observe substantive efficiency gains.
>
> In Table A1, we use the large-scale controlled evaluation setting described in Section 4.3 and sample each of Claude, Gemini, and GPT-5 5 times, in keeping with the Conductor’s allowable workflows of 5 steps. We used MMLU as a representative task. We report cost in dollar-average per sample, and cost-adjusted performance as performance / average cost in cents. We see that the Conductor outperforms the frontier models under consensus and self-reflection inference-time scaling, agreeing with our expectation that through harnessing the varying capabilities across the agent pool, the Conductor is able to access substantive performance gains beyond any single agent. Second, we furthermore see that the Conductor is a highly efficient framework, with token usage per sample and average cost per sample substantially lower than the inference-time sampling strategies. This again agrees with our prior intuition that, as the Conductor does not rely on hand-designed or fixed sampling strategies, it can instead adaptively assemble the agents according to the task at hand.
>
> Table A1
>
> | Model                          | Performance | Token Usage | Cost      | Cost-adjusted performance |
> |--------------------------------|-------------|-------------|-----------|---------------------------|
> | Claude Sonnet 4 5x consensus   | 91          | 1412.8      | 0.021192  | 42.94073235               |
> | Claude Sonnet 4 5x reflect     | 90.66       | 2516.992    | 0.02080128| 43.58385638               |
> | Gemini 2.5 Pro 5x consensus    | 91.6        | 1658.4      | 0.016584  | 55.23396044               |
> | Gemini 2.5 Pro 5x reflect      | 88.33       | 2919.776    | 0.01675976| 52.70361867               |
> | GPT 5 5x consensus             | 91.3        | 1376.3      | 0.013763  | 66.33728112               |
> | GPT 5 5x reflect               | 91.79       | 2457.132    | 0.01424907| 64.41823923               |
> | Conductor                      | 93.14       | 735.2       | 0.009     | 103.4888889               |
>
>
> In Table A2 below, we also report the performance and efficiency results for Conductor in comparison with Smoothie, RouterDC, MoA, and MASRouter. We obtain these results using our mixture of 4 training tasks, MMLU, RLPR, Math500, and LiveCodeBench. We see, in line with our expectation, that the substantially lower API calls of the Conductor similarly translate into the second-lowest token usage (behind only RouterDC, which selects only a single model). Conductor’s average cost per sample is behind only RouterDC and MASRouter, models for which it offers a roughly 40% relative improvement across the spread of 4 tasks. Relative to MoA, the next best performing baseline, the Conductor is over twice as cost-efficient while boosting performance by almost 20% in relative terms.
>
> Table A2
>
> | Model   | Performance | Token Usage | Cost |
> |----------------|---------|----------|------------------|
> | MoA            | 62.13 | 11203 |  $0.048554  |
> | Smoothie    | 56.48  |9909   | $0.039291  |
> | RDC            | 52.41  | 840    | $0.005613 |
> | MasRouter  | 56.89 | 4970   | $0.01345   |
> | Conductor   | 72.35 | 1820   | $0.02384   |
>
> Following the reviewer's comment, **we have added these additional performance-efficiency results to Appendix B of our revised manuscript**.
>
> [1] Wang, Xuezhi, et al. "Self-consistency improves chain of thought reasoning in language models." International Conference on Learning Representations (ICLR) (2023)

---

> ### Author Response · Authors · 2025-11-20
> **Response to reviewer (4/6)**
>
> > **b** There is no efficiency benefits for this because everything is sequential and one model has to wait for last model to finish (correct me if I misunderstood). This limits the contribution of the Conductor.
>
> We would like to clarify that our framework fully allows the Conductor to specify arbitrary tree-like coordination strategies that are non-sequential, and can thus be executed in parallel without waiting for prior models. A concrete example of the Conductor learning to harness this capability is provided in Figure 19 (Appendix F), where we record a frequent observation where the Conductor learns to use a tree topology with the first two agents queried in parallel, with subtasks that can be completed independently. **Following the reviewer’s comment, in order to avoid future potential misunderstanding with our reader, we now explicitly mention this important property when describing our method in Section 3 of the latest revision.**.
>
>
> > **Question 1:** How sensitive is the Conductor’s learning to the choice of reward granularity (binary correctness vs. partial credit or verification-based rewards)? Would denser reward shaping improve stability or lead to different emergent strategies? What reward strategies have you guys tried? I am generally really curious about this.
>
> We thank the reviewer for their interest in the empirical history of what shaped our designs. In the early stages of our development, we experimented with a few different reward structures and indeed found some interesting behaviors emerging, for which we have included the main findings below. In summary, we found that the Conductor is fairly robust across different choices of sensible reward designs. Nonetheless, we also found that complex strategies are unnecessary for optimal performance, with the powerful coordination strategies reported in the main body of the manuscript emerging readily from our end-to-end optimization of the simple, sparse reward structure detailed in Section 3.1. **Following the reviewer’s interest, we’ve added an  extended discussion in Appendix G describing some of these emergent behaviors**, which we include here below for easier reference:
>
> ---
> Geometric rewards: We experimented with a simple geometric reward structure defined as $reward = format \times correctness$ where format = 1 iff the Conductor completion was correctly formatted and 0 otherwise. This geometric structure is equivalent to removing partial rewards for well-formatted but incorrect completions. In line with expectations, such a reward structure still performed well and led to the Conductor learning performant coordination strategies, but was abandoned in favor of the existing setup with partial rewards for well-formatted completions. The geometric reward produced a lower diversity in coordination strategies (decreased exploration of the coordination space) and was more likely to lead to suboptimal local optima, for example, by over-relying on less powerful open-weight models. We also found, qualitatively, that the Conductor was much more likely to mimic the few-shot examples provided as part of the Conductor prompt, which is intuitive given that they represent sure ways to start generating non-zero rewards by having correct formatting. Again, this hurt the exploration of the coordination space. Empirically, we nonetheless saw fairly strong performance (in the range of 3-5% worse than our final model across the board), as the Conductor still discovered the most powerful agents and how to coordinate them, just often taking longer to discover them and occasionally settling in suboptimal minima.
>
> Non-chain-bonus: We observe that the Conductor discovers tree topologies (two or three agents work independently, and a final model has access to all the previous agents' work) and chains (each agent has access to all the work done before it) as the optimal collaborative topologies. This is, in a sense, unsurprising, as trees and chains are intuitively powerful, allowing successive models to build off of the work done by other agents. However, we nonetheless experimented with providing small rewards to the Conductor for exploring alternate collaborative topologies to see if powerful, less-intuitive topologies would emerge. This led to the Conductor discovering an interesting topology in which the Conductor used all 5 workflow steps to create two small, independent working groups (2 models per group) and then a final model supervising both groups. We note that exploring further collaborative topologies will likely necessitate raising the workflow steps to 10 or beyond and widening the worker pool, and we plan to pursue this interesting line of research in the future.

---

> ### Author Response · Authors · 2025-11-20
> **Response to reviewer (5/6)**
>
> Cost-bonus: We also experimented with cost bonuses, i.e, continuous rewards depending on the total cost of the coordination strategy as measured by the input/output token costs of work done by the selected agents. This reward provided several axes for altering the emergent strategies discovered by the Conductor. One intuitive consequence was the Conductor relying on fewer models on average (the prevalence of workflows with 2 agents or just a single agent increased substantially), and furthermore relying on the open-source models to a much higher degree. One example of this is the Conductor utilizing open-source models to do the majority of the work – including planning, drafting solutions, and performing the actual solving – before having a closed-source model act as a final verifier. An additional interesting behavior that we observed in the early stages of training was the Conductor trying to take on more of a solving role itself, where it tried to partially solve the question itself or give highly directed possible strategies to the agent models, thereby redistributing some of the work off of the agents in order to save cost.
>
> Recursion transition rewards: In our recursive setup (Section 3.2), we also experimented with a transition reward structure described as follows:
> $$\text{if WR, then } r = 1$$
> $$\text{if RW, then } r = -1$$
> $$\text{if RR, then } r = 0.5$$
> $$\text{if WW, then } r = 0$$
> where WR denotes wrong in the initial round and right in recursive round (I.e the conductor accurately rectified a coordination strategy that produced an incorrect answer), RW denotes right in the initial round and wrong in the recursion round (ie the Conductor revised a correct strategy to an incorrect one), WW denotes wrong in both rounds and RR denotes right in both rounds. Hence, this reward structure was designed to maximally reward the rectification of incorrect answers and maximally punish the incorrect revision of already-correct answers.
>
> This transition reward structure produced the intuitive behavior where the recursive Conductor became overly conservative with regard to calling itself to revise a final solution. The rate at which the Conductor revised previous coordination strategies fell to below 20%, where for those problems in which the Conductor chose to revise the strategy, the Conductor would typically recall the same models to check over their work once more. This represented a kind of collapse in recursion, since the very point of recursion (iterative adaptation of strategies on the fly) was nullified, as the recursive strategies just collapsed back to the initial coordination strategies, which caused recursive performance to be unable to surpass generic Conductor performance.
>
> Agent feedback: We also experimented with providing rewards to the Conductor issued by agents, i.e, leveraging the more powerful agent LLMs as judges, where the agents would issue small negative rewards if their assigned subtasks were poorly thought-out. In this setup, however, it was difficult to discern much of a difference in Conductor behavior, mostly because poorly thought-out subtasks happened so rarely, and there already existed a powerful mechanism for punishing poorly thought-out coordination strategies – i.e, the correctness reward itself. Nonetheless, we remain curious about exploring this reward structure in the future, and are optimistic that there may be new and more powerful coordination strategies discoverable by leveraging feedback from the agents as partial rewards.
>
> ---
> We hope this new discussion we added following the reviewer's interest will provide valuable and informative insights to future work and extension in this domain.

---

> ### Author Response · Authors · 2025-11-20
> **Response to reviewer (6/6)**
>
> > **Question 2** What are the different modes of orchestration the Conductor learned? Did you guys do some simple categorization?
>
> One of the most interesting things, in our opinion, about the Conductor is that we continue to observe new modes of orchestration as we keep applying it to new tasks. Explicit categorization was difficult given the wide range of coordination strategies we observed, but we did take note of the most frequent kinds of strategies following specific patterns. These were:
>
> 1) Sequential coordination with a high-level planner, followed by an executor, usually with additional models in between acting as refiners and checkers (exemplified in Figures 23 and 24).
> 2) Tree coordination, where the first agents work independently and an additional agent aggregates the independent work (Figure 19).
> 3) Tree coordination, where the conductor recognizes the question depends purely on factual recall, and thereby does not necessitate any agent-to-agent collaboration (Figure 27).
> 4) Sequential coordination with a highly logical reasoning process, where each agent proceeds step-by-step to decompose a challenging multi-step problem (Figure 20).
> 5) Conductor task abdication (one of our personal favorites), where the Conductor passes its own role onto one of the more powerful agent models, e.g, asking GPT-5 to come up with subtasks and direct the other models what to do (Figure 28)
>
> Following the reviewer's question, **we extended Appendix F to specifically list and categorize these different modes**, which we hope will help provide further transparency into our model's inner workings for interested readers.

---

### Official Review · Reviewer_5Z3X · 2025-11-01

**Soundness:** 3
**Presentation:** 3
**Contribution:** 4
**Rating:** 8
**Confidence:** 4

**Summary:**

This paper introduces a conductor model trained with RL to "orchestrate" between different underlying LLMs to perform work. The conductor attains SOTA on reasoning benchmarks and outperforms all component models. The authors also enable online recursive adaptation of the conductor agent. This unlocks a new form of dynamic test-time compute scaling.

**Strengths:**

This is an excellent paper. The training setup is sound and the conductor is trained end-to-end with RL. This strategy unlocks new SoTAs on top reasoning benchmarks. The training setup also clearly observes novel strategies from the conductor (e.g., having sub-LLMs validate each other's outputs and plan with each other). There are a number of ablations to prove robustness and follow-ups to results, and many of them have interesting results of their own (e.g., showing the impact of spending more agents on harder tasks). Figures are clean. Examples are thorough.

**Weaknesses:**

It is unclear if these techniques improve performance compared to other test-time-compute strategies (e.g., pass@k, best-of-N, consensus, or other heavily prompted setups to increase test-time-compute put into solving the same problem). Because the incremental lift from this strategy seems relatively small, this paper would be significantly strengthened by including comparisons to other test-time-compute strategies to show that it's worth it compared to e.g., pass@k/BoN/cons/prompting/etc.

Moreover, it would be useful to understand "cost-normalized" performance; quantitatively, much more latency and cost is incurred from using this strategy and is it worth the performance gained? There is a plot with number of agent calls but latency in terms of time/tokens and cost in terms of actual API pricing or equivalent would be much more interpretable here to baseline the tradeoff needed to get the performance wins.

**Questions:**

What is the "non-trained" baseline (e.g., BoN/pass@)? What is cost-normalized performance?

---

> ### Author Response · Authors · 2025-11-20
> **Response to reviewer (1/2)**
>
> >**Weakness 1:** It is unclear if these techniques improve performance compared to other test-time-compute strategies. This paper would be significantly strengthened by including comparisons to other test-time-compute strategies to show that it's worth it compared to e.g., pass@k/BoN/cons/prompting/etc.
>
> We thank the reviewer for the suggestion regarding including inference-time frameworks, and agree that this will significantly help us to demonstrate the substantiveness of our performance gains. **Following the reviewer's suggestion, we ran additional evaluations of GPT-5, Gemini 2.5 Pro, and Claude Sonnet 4 using consensus inference-time scaling** [1]. We used the evaluation setup described in Section 4.1, and based the consensus on 5 samples from each model, in line with the Conductor’s own max workflow sizes of 5 steps. We used MMLU as the representative task. Alongside performance, we additionally report average token usage per sample, average cost per sample, and cost-adjusted performance (performance / average cost per sample in cents), in order to provide a full picture of the Conductor’s performance and efficiency benefits in comparison with test-time compute strategies. We also included in these results our 5x self-reflection baseline as a further inference-time scaling prompting strategy, and added the same token usage and cost information. We present the results in Table A1 below, where we see that the Conductor improves over these inference-time strategies through harnessing the collective capabilities across its available workers.
>
> First, we see that the Conductor outperforms each frontier model when using a 5x consensus and 5x self-reflect inference-time scaling strategy. This agrees with our expectation, since the Conductor is able to surpass any single agent’s capabilities through harnessing the complementary skills of the entire pool of agents. Second, we see that consensus and self-reflection scaling are only able to push performance by fractions of a percent. This helps to highlight the substantiveness of our Conductor’s performance gains, as we see that pushing the performance of these powerful frontier models is highly non-trivial. Third, we furthermore see that the Conductor is a highly efficient framework, with average token usage and average cost per sample roughly half that of the inference-time scaling frameworks. This also agrees with our expectation, given that the Conductor adaptively assembles its constituent workers to solve a task, and so doesn’t rely on any fixed rules regarding inference-time frameworks, as consensus does.
>
> These findings thus show that the Conductor, through its powerful coordination strategies that compose the varying skills of different models, offers an impactful way to access higher performance and efficiency not available to existing inference-time frameworks. **We’ve added these additional results to Appendix B of our updated manuscript**.
>
>
> Table A1
>
> | Model                          | Performance | Token Usage | Cost      | Cost-adjusted performance |
> |--------------------------------|-------------|-------------|-----------|---------------------------|
> | Claude Sonnet 4 5x consensus   | 91          | 1412.8      | 0.021192  | 42.94073235               |
> | Claude Sonnet 4 5x reflect     | 90.66       | 2516.992    | 0.02080128| 43.58385638               |
> | Gemini 2.5 Pro 5x consensus    | 91.6        | 1658.4      | 0.016584  | 55.23396044               |
> | Gemini 2.5 Pro 5x reflect      | 88.33       | 2919.776    | 0.01675976| 52.70361867               |
> | GPT 5 5x consensus             | 91.3        | 1376.3      | 0.013763  | 66.33728112               |
> | GPT 5 5x reflect               | 91.79       | 2457.132    | 0.01424907| 64.41823923               |
> | Conductor                      | 93.14       | 735.2       | 0.009     | 103.4888889               |
>
>
> [1] Wang, Xuezhi, et al. "Self-consistency improves chain of thought reasoning in language models." International Conference on Learning Representations (ICLR) (2023)

---

> ### Author Response · Authors · 2025-11-20
> **Response to reviewer (2/2)**
>
> > **Weakness 2: it would be useful to understand "cost-normalized" performance**; quantitatively, much more latency and cost is incurred from using this strategy and is it worth the performance gained? There is a plot with number of agent calls but latency in terms of time/tokens and cost in terms of actual API pricing or equivalent would be much more interpretable here to baseline the tradeoff needed to get the performance wins.
>
> We thank the reviewer for the suggestion and agree that a token / cost analysis would improve interpretability and help demonstrate the improved efficiency of our framework. **Following the reviewer’s suggestion, we conducted additional experiments on token usage, average cost, and cost-adjusted performance of the Conductor in relation to the multi-agent baselines**.  We present the updated performance / efficiency analysis below in Table A2, where we see that, in line with our Figure 5, the Conductor offers improved cost-normalized performance versus the existing multi-agent baselines.
>
> In Table A2, we report the performance and efficiency results for Conductor in comparison with Smoothie, RouterDC, Smoothie, MoA, and MASRouter. We obtain the results using our mixture of 4 training tasks, MMLU, RLPR, Math500, and LiveCodeBench. We see, in line with our expectation, that the substantially lower API calls of the Conductor similarly translate into second-lowest token usage (behind only RouterDC, which selects only a single model). Conductor’s average cost per sample is behind only RouterDC and MASRouter, models for which it offers a roughly 40% relative improvement across the spread of 4 tasks. Relative to MoA, the next best performing baseline, the Conductor is over twice as cost-efficient while boosting performance by almost 20% relatively.
>
> Table A2
>
> | Model   | Performance | Token Usage | Cost |
> |----------------|---------|----------|------------------|
> | MoA            | 62.13 | 11203 |  $0.048554  |
> | Smoothie    | 56.48  |9909   | $0.039291  |
> | RDC            | 52.41  | 840    | $0.005613 |
> | MasRouter  | 56.89 | 4970   | $0.01345   |
> | Conductor   | 72.35 | 1820   | $0.02384   |
>
> **We’ve added these additional results to Appendix B of our updated manuscript**.
>
> >**Q1:** What is the "non-trained" baseline (e.g., BoN/pass@)? What is cost-normalized performance?
>
> Alongside our 5x self-reflection inference-time baselines, **we've added evaluation with consensus inference-time scaling**, providing performance and cost analysis to demonstrate the improved cost-adjusted performance and overall efficiency of our Conductor. Our experiments found that, in line with our expectation, these inference-time strategies struggle to deliver substantive performance gains as they are fundamentally bottlenecked by the capability of that single agent. Our Conductor, by contrast, through composing and leveraging the differing capabilities across a pool of agents, is able to deliver marked performance gains, which are otherwise difficult to access through inference-time sampling or prompting strategies.
>
> To best assess cost-normalized performance, **we've provided token usage and average cost per sample across single-agent inference-time frameworks and multi-agent baselines in comparison with our Conductor in Tables A1 and A2**. In both settings, we find that the Conductor surpasses baseline models in performance through its powerful coordination strategies that best leverage the capabilities of the agents. At the same time, the Conductor remains highly efficient through its adaptive, sample-dependent strategies that do not rely on any fixed or manually designed agent scaffolds or sampling strategies. Altogether, these findings demonstrate the Conductor offers substantive improvements in cost-normalized performance.

---

### Official Review · Reviewer_TwCE · 2025-11-01

**Soundness:** 3
**Presentation:** 1
**Contribution:** 2
**Rating:** 2
**Confidence:** 3

**Summary:**

The paper proposes training a small (7B) Conductor model using reinforcement learning (RL) to orchestrate multiple strong LLMs for complex reasoning tasks. The model is trained using GRPO for 200 iterations. The experiments demonstrate that the Conductor achieves better performance on these reasoning tasks compared to any single model. Overall, the experimental results largely support the claim that the Conductor model improves reasoning performance in practice, although the margin of improvement is not always substantial.

**Strengths:**

- The core idea of using reinforcement learning to train a dedicated task planner/assigner (the "Conductor") is a potentially interesting direction to combine different models’ strengths.
- The experimental results demonstrate that this approach is effective, showing performance gains over individual state-of-the-art models.

**Weaknesses:**

- **The presentation has significant flaws.** Some expressions are imprecise, lack formal explanations, or are unsuitable for an academic paper. For instance, the GRPO formula in Equation (1) appears to be incorrect, as it seems to be missing the clipping mechanism characteristic of PPO-style algorithms. Also, the paper frequently uses the term "agentic workflows" without providing a formal definition within the context of this study. This term is often over-used, and the authors need to clarify precisely what it entails in their framework. Several other key terms are left undefined or used imprecisely. For example:
    - What do the authors mean by the "latent capability" of LLMs (line 37)?
    - What constitutes the "unconstrained" setting for evaluation (line 267)?
- **The motivation for training a separate Conductor model is not fully convincing given the experimental results.** While Table 1 and Figure 4 show that the Conductor improves upon the best single agent (model), the performance foundation clearly comes from the powerful frontier models it orchestrates (e.g., GPT-5 achieves 90.8 on unseen task AIME25, while Conductor reaches 93.3). The marginal gain seems small. This raises a critical missing baseline: what is the performance of a strong model (like GPT-5 or Gemini 2.5 Pro) when simply prompted to act as the task planner and assigner? Furthermore, the results for other multi-agent baselines in Figure 4 are confusing. Why do established frameworks like MoA and MASRouter show results inferior to a single-agent Gemini 2.5 Pro? Were these baselines tested with the same powerful set of worker models as the Conductor? This needs clarification to fairly assess the Conductor's contribution.

**Questions:**

- The reward mechanism is described at a high level. Could you elaborate on the credit assignment? Is the final reward applied to the entire sequence of tokens generated by the Conductor? How does this scalar reward effectively train the complex, multi-step workflow generation?
- The premise of the paper is that different models have specialized, complementary skills. Could you provide concrete examples of this from your experiments? For instance, are there cases where a generally "weaker" model (like Qwen3-32B) correctly solves a sub-task that a "stronger" model (like GPT-5) fails at, demonstrating true complementary specialization?
- How much more computational cost (e.g., inference rounds or total tokens) does the Conductor framework introduce compared to the single-model / single-agent scenarios? A clear analysis of the performance-cost trade-off is needed.

---

> ### Author Response · Authors · 2025-11-20
> **Response to reviewer (1/6)**
>
> > **Weakness 1**. The presentation has significant flaws. Some expressions are imprecise, lack formal explanations, or are unsuitable for an academic paper.
>
> We thank the reviewer for identifying areas of improvement in the paper's presentation. Following their feedback, we made several edits in the latest revision of our paper to make our definitions and notation clearer. Specifically, regarding the points noted by the reviewer:
>
> > **a.** For instance, the GRPO formula in Equation (1) appears to be incorrect, as it seems to be missing the clipping mechanism characteristic of PPO-style algorithms.
>
> The reason equation 1 omitted the clipping term is that the Conductor is trained using a single optimization step per rollout (as in DeepSeek R1 [1]), in which case both the clipped and unclipped objectives are equivalent. The alternative unclipped notation for this special case is also employed in prior work [2, 3], including Appendix A of the DeeSeekMath paper [4].
>
> Following the reviewer's comment, **we changed Equation (1) to the full clip version and added a better explanation of our hyperparameters** in Section 2, to avoid potential misunderstandings.
>
> > **b.** Also, the paper frequently uses the term "agentic workflows" without providing a formal definition within the context of this study. This term is often over-used, and the authors need to clarify precisely what it entails in their framework.
>
> In the first paragraph of Section 3.1, we defined an agentic workflow as a sequence of "workflow steps", each comprising a natural language substask, an integer id corresponding to the assigned worker agent responsible for performing that subtask, and an access list indexing which previous subtask solutions to include in the worker’s context. Following the reviewer's comment, we realize that this important definition should be better highlighted, as we used it several times in the paper. Following the reviewer's comment, **we have now placed this definition in a separate, formal "Definition" block and added explicit references to it throughout the paper**.
>
> > **c.** Several other key terms are left undefined or used imprecisely. For example:
> What do the authors mean by the "latent capability" of LLMs (line 37)?
> What constitutes the "unconstrained" setting for evaluation (line 267)?
>
> By "latent capabilities" of LLMs, we meant the improved performance and in-context learning capabilities unlocked by modern prompting and context manipulation techniques. To better contextualize this term, **we have now added explicit citations to prior seminal work in these domains after our use of the "latent capabilities" term**, including [5, 6]. By "uncostrained", we refer to the definition we gave in the first paragraph of Appendix A.3, where we detail how we prompt each agent without output token limits, and compare with the best reported results across both our re-implementations and online benchmarks. Following the reviewer's question, **we now added an explicit reference to Appendix A.3 in line 267**, to make the connection to our definition clearer.
>
> We hope these modifications, based on the reviewer's feedback, have managed to improve the presentation and clarity of the latest submitted revision. We hope the reviewer will not hesitate to let us know in case they believe there is any other significant aspect of our text and exposition that we should work to improve.
>
> [1] Guo, Daya, et al. "Deepseek-r1: Incentivizing reasoning capability in llms via reinforcement learning." arXiv preprint arXiv:2501.12948 (2025).
>
> [2] Cetin, Edoardo, Tianyu Zhao, and Yujin Tang. "Reinforcement Learning Teachers of Test Time Scaling." arXiv preprint arXiv:2506.08388 (2025).
>
> [3] Li, Gang, et al. "DisCO: Reinforcing Large Reasoning Models with Discriminative Constrained Optimization." arXiv preprint arXiv:2505.12366 (2025).
>
> [4] Shao, Zhihong, et al. "Deepseekmath: Pushing the limits of mathematical reasoning in open language models." arXiv preprint arXiv:2402.03300 (2024).
>
> [5] Liu, Pengfei, et al. "Pre-train, prompt, and predict: A systematic survey of prompting methods in natural language processing." ACM computing surveys 55.9 (2023): 1-35.
>
> [6] Brown, Tom, et al. "Language models are few-shot learners." Advances in neural information processing systems 33 (2020): 1877-1901.

---

> ### Author Response · Authors · 2025-11-20
> **Response to reviewer (2/6)**
>
> > **Weakness 2**: The motivation for training a separate Conductor model is not fully convincing given the experimental results.\
> > **a**  While Table 1 and Figure 4 show that the Conductor improves upon the best single agent (model), the performance foundation clearly comes from the powerful frontier models it orchestrates (e.g., GPT-5 achieves 90.8 on unseen task AIME25, while Conductor reaches 93.3). The marginal gain seems small.
>
> As the main results provided in Table 1 and Figure 4, for which the Conductor achieves state-of-the-art results, do rely on powerful frontier models to orchestrate, we can understand the reviewer's perspective. Thus, in our updated revision, **we have added an extended discussion to Sections 4.2, 4.4, and Appendix A, which we hope will better highlight two aspects regarding the nature of our evaluated tasks and the meaningfulness of the Conductor's capabilities**:
>
> 1. Highly competitive reasoning benchmarks, such as AIME, tend to have a long-tailed distribution of difficulty [1] where breakthroughs in a small subset of particularly challenging problems could be representative of entire generational improvements in LLM reasoning. For example, the difference between two generations of GPT reasoning models, from GPT-o3 and GPT-5, is 3.3% on AIME25 [2] and 2.7% on GPQA-Diamond [3] in absolute percentage terms. Moreover, the Conductor framework not only yields improvements in a single domain but across a diverse range of such highly challenging benchmarks, across math, coding, and natural science, something we believe particularly validates the meaningfulness of its advancements beyond the performance foundation of frontier models.
>
> 2. In Figure 6, we also provide results with the performance of the Conductor when used to orchestrate exclusively open models that are individually much weaker than any single closed-source model we considered. Nonetheless, even in this constrained setting, we find the Conductor can still effectively prompt engineering and coordinate its less powerful agents, raising performance by almost 10 points to even exceed Claude-4 Sonnet on LiveCodeBench. We find this result particularly indicative of the fact that the Conductor is not exclusively reliant on the performance foundation of frontier models, displaying even larger absolute gains when using a foundation with a larger room for improvement.
>
> [1] Xu, Chengyin, et al. "Unveiling downstream performance scaling of llms: A clustering-based perspective." arXiv preprint arXiv:2502.17262 (2025).
>
> [2] “AIME 2025 Leaderboard.” Kaggle, https://www.kaggle.com/benchmarks/open-benchmarks/aime-2025.
>
> [3] “GPQA Diamond Benchmark Leaderboard.” Artificial Analysis, https://shorturl.at/eLHUj

---

> ### Author Response · Authors · 2025-11-20
> **Response to reviewer (3/6)**
>
> > **b**. What is the performance of a strong model (like GPT-5 or Gemini 2.5 Pro) when simply prompted to act as the task planner and assigner?
>
> Following the reviewer’s suggestion, **we set up three additional baselines**. The first, labelled "**GPT-5 conduct 7 models**", tasks GPT-5 to act as the conductor, allowing it to design coordination strategies of up to 5 workflow steps with access to the same set of 7 workers as our model (GPT 5, Gemini 2.5 Pro, Claude Sonnet 4, R1-distill-Qwen32B, Qwen3-32B (thinking), and Qwen3-32B (nonthinking)). The second and third labelled "**GPT-5 conduct 3 models**" and "**Gemini conduct 3 models**",  further restrict the agent pool to only the best-performing large frontier models (GPT 5, Gemini 2.5 Pro, Claude Sonnet 4), as we noticed an over-reliance on open-source models in highly suboptimal ways following preliminary experiments with "GPT-5 conduct 7 models" (for example by asking R1-distill-Qwen-32B to act as a final checker before returning the solution to the user, often resulting in solution formatting failures).
>
> To make sure these baselines performed at the best of their ability, we also incorporated an automatic resampling strategy whenever GPT-5 or Gemini produced a format failure (with unlimited retry attempts) and doubled the conductor's original output token limit in order to make better use of their reasoning capabilities. **We added these results and an extended analysis to Table 11 of the latest revision**, which we summarize below:
>
> Table B
>
> | Model                     | LCB   | AIME | BigCodeBench | GPQA-D | Avg.    |
> |---------------------------|-------|------|--------------|--------|---------|
> | GPT-5 conduct 7 models   | 50.86 | 76.67 | 34.5       | 77.78 | 59.9525  |
> | GPT-5 conduct 3 models            | 67.43 | 93.3 | 33.1         | 86.36  | 70.0475 |
> | Gemini 2.5 Pro conduct 3 models   | 70.29 | 93.3 | 35.13        | 87.62  | 71.585  |
> | Conductor                 | 83.93 | 93.3 | 37.86        | 87.5   | 75.6475 |
>
> First, we note that the Conductor expectedly maintains its superior performance across tasks, confirming the effectiveness of our training strategy. In particular, both GPT-5 and Gemini appear over-reliant on an initial pool of models based on their prior biases, which fails to match actual downstream performance. A prime example of this is that the baselines fail to understand that Claude is less capable at LiveCodeBench or GPT-5 is less capable at BigCodeBench, leading to deteriorated performance without an effective feedback mechanism to adjust their knowledge misconceptions, as provided by the Conductor's training phase. Second, we interestingly find that, despite having never been trained on these tasks, both "GPT-5 conduct 3 models" and "Gemini conduct 3 models" outperform their constituent agents (GPT5 and Gemini) across numerous tasks, and their performance is also visibly superior to our much smaller base 7B Qwen model before any training. We believe this provides further evidence validating the Conductor's underlying hypothesis that powerful LLMs are inherently suitable to act as effective meta-orchestrators, highlighting the potential of harnessing future and or larger base models as a simple direction to scale our new framework down the line.

---

> ### Author Response · Authors · 2025-11-20
> **Response to reviewer (4/6)**
>
> >**c**: Why do established frameworks like MoA and MASRouter show results inferior to a single-agent Gemini 2.5 Pro? Were these baselines tested with the same powerful set of worker models as the Conductor?
>
> First, as described in the first paragraph of Section 4.3, all multi-agent baselines were trained and tested with the exact same set of agents as the Conductor. These were GPT-5, Gemini 2.5 Pro, Claude Sonnet 4, Qwen-32B (thinking), Qwen-32B (nonthinking), R1-distill-Qwen-32B, and gemma-3-27b-it. **Following the reviewer's concern, we’ve updated our manuscript to better clarify and emphasize this important point in future revisions.**
>
> Second, following the reviewer’s questions on MoA and MASRouter underperforming frontier models in certain tasks, **we’ve added an extended discussion complete with examples from the evaluation logs in Section 4.3 and Appendix B**, which we summarize in the following paragraphs:
>
>  As the reviewer correctly noted, on LiveCodeBench, the MoA baseline underperformed GPT-5, failing to leverage GPT-5’s impressive capabilities. Examining the evaluation logs reveals that the drop in performance is explained by MoA suboptimally using the candidate solutions of less capable models to inform the final response, or often being misled by the incorrect solutions of other models. Indeed, we see that for other tasks where models are closer in capability, MoA performed better, but for LiveCodeBench, where there is a high variance in performance, MoA struggled to discern the correct answer or combination of answers among the 7 candidate responses. **We note that this finding echoes the result obtained in Table B suggested by the reviewer**, showing that open-weight models can degrade the performance of the frontier models when combined suboptimally, with particularly marked performance drops in LiveCodeBench. We posit that this reveals a property of MoA, for which performance depends in large part on the ability to discern correct from incorrect, which becomes increasingly challenging in tasks with very large solution spaces (for example, writing optimized code).
>
> Regarding MASRouter, we instead find that this baseline relies heavily on human-engineered scaffolding techniques, which require careful placements of selected models for specific roles. Therefore, when asked to make many specific decisions about model and role selection, MASRouter can struggle to determine which models are best suited for each task. This issue is especially evident when MASRouter is exposed to our wide pool of worker agents, where discovering the optimal combinations of agents and allocations of roles is challenging. Additionally, the manually designed setup of MASRouter forces it to rely on fixed prompt templates and scaffolds to direct models to solve problems, with limited generality beyond the domains it was designed for. By contrast, the Conductor uses no human-designed fixed prompts, can leverage LLM’s strong generalization properties, and learn how to write custom focused subtasks that best leverage their capabilities for each question.
>
> > **Question 1.** The reward mechanism is described at a high level. Could you elaborate on the credit assignment? Is the final reward applied to the entire sequence of tokens generated by the Conductor? How does this scalar reward effectively train the complex, multi-step workflow generation?
>
> The Conductor is itself a language model trained to output the full agentic workflows in natural language in a single generation. Thus, as the reviewer correctly suggested, the final binary task reward is used to optimize the entire sequence of tokens end-to-end. We find that this simple strategy yields the emergence of problem breakdowns and prompt-engineered subtasks that match each worker's strengths. We believe these findings are remarkably in line with ones from the DeepSeek R1 paper, showing that simplicity and end-to-end optimization can often constitute a great recipe for training language models towards challenging objectives.
>
> Following the reviewer's question, **we made edits to the text in Section 3.1 to explicitly note the simplicity of the methodology and remark on the connection to the DeepSeek R1 results to better clarify our approach**.

---

> ### Author Response · Authors · 2025-11-20
> **Response to reviewer (5/6)**
>
> > **Question 2.** The premise of the paper is that different models have specialized, complementary skills. Could you provide concrete examples of this from your experiments? For instance, are there cases where a generally "weaker" model (like Qwen3-32B) correctly solves a sub-task that a "stronger" model (like GPT-5) fails at, demonstrating true complementary specialization?
>
> We frequently observed that different models display specialized complementary skills  both at a “task level”, if the Conductor uses a model as the main driver of performance, but also at a more granular “subtask level” when dealing with only specific aspects of the input question. Our observations also include several instances where a generally "weaker" model can correctly solve a task or sub-task that a "stronger" model fails. Following the reviewer's interest, **we’ve added additional qualitative examples (Figures 23, 24, 25, and 27) to Appendix F and discussion to Appendix A**, demonstrating and comparing this behavior across the aforementioned axes, which we summarize below:
>
> ---
> At the global "task level", differing closed-source models visibly excel in differing tasks. Immediate examples of this are GPT-5’s strong performance in math and competitive coding (seen in AIME and LiveCodeBench), while Gemini excels in scientific reasoning (GPQA-Diamond). Claude Sonnet 4 struggles at competitive coding (relatively weak in LiveCodeBench), but is one of the dominant models at code generation with diverse function calling (BigCodeBench). This specialization is also prevalent at the granular "sub-task" level, where we often find the Conductor learns that different models are most useful as "planners" or "writers" to answer particular kinds of questions.  For instance, our SOTA performance in LiveCodeBench leverages Gemini 2.5 Pro and Claude Sonnet 4 working together as high-level planners and only later employs GPT-5  to write the final optimized code, which far outperformed alternate strategies using GPT-5 in a planning role that the Conductor attempted at early training iterations.
>
> Furthermore, regarding ‘weaker’ open-source models, we indeed observed concrete instances where these models could fill roles and solve particular questions that their closed-source counterparts failed at. We note this was more prevalent at a subtask level, where using GPT-5 as the final validator for several BigCodeBench questions would fail to adhere to the benchmark's strict formatting requirements, disregarded by this agent. In these instances, simply switching away to the much smaller Qwen3-32 or DeepSeek as final validators would allow the Conductor to succeed. While expectedly less common, we also found examples at the global "task level", with most such cases in RLPR and MMLU, with questions outside of traditional mathematical reasoning, such as:
>
> “Four men who would become fathers were in a hospital waiting room while their wives were going to give birth to their babies. The nurse arrived and proudly said to the first man, \"Congratulations, sir...[truncated]...d never have taken that job at 7-Eleven.\" At least how many babies would be born according to the story?”
>
> For this particular MMLU question, we interestingly found that GPT-5 failed to produce correct responses across repeated trials, failing to match the level consistently attained by directly prompting Gemma 27B.
>
> ---
>
> We hope this new extended discussion and analysis that the reviewer suggested will be a valuable addition to our paper and provide a stronger motivation for our new approach.

---

> ### Author Response · Authors · 2025-11-20
> **Response to reviewer (6/6)**
>
> >**Q3:** How much more computational cost (e.g., inference rounds or total tokens) does the Conductor framework introduce compared to the single-model / single-agent scenarios? A clear analysis of the performance-cost trade-off is needed.
>
> We thank the reviewer for raising this important point, which we realize could be more comprehensively studied in our initial revision.. **Following the suggestion, we’ve run additional single-agent baselines using Claude Sonnet 4, Gemini 2.5 Pro, and GPT-5 under a consensus [1] inference-time strategy in order to provide a more comprehensive picture of the Conductor’s performance-cost tradeoff**. We use the large-scale controlled evaluation setting described in Section 4.3 and sample each model 5 times, in keeping with the Conductor’s allowable workflows of 5 steps. We used MMLU as a representative task to conduct the analysis and report performance, average token usage per sample, average dollar cost per sample, and cost-adjusted performance (calculated as performance / average cost in cents) for all models. Furthermore, we also include our 5x self-reflection baseline as a further inference-time scaling strategy for these results, and add the same token usage and cost information in Table A1 below.
>
>
> First, we see that the Conductor outperforms the frontier models under consensus and self-reflection inference-time scaling, in line with our empirical findings that, through harnessing the varying capabilities across the agent pool, the Conductor attains substantive performance gains beyond any single agent. Second, we see that the Conductor is a highly efficient framework, with token usage per sample and average cost per sample substantially lower than the  inference-time sampling strategies. This is in line with our intuition, as the Conductor does not rely on hand-designed or fixed sampling strategies, and instead adaptively assembles the agents according to the task at hand. Third, these findings echo findings in prior work, highlighting that generating performance gains through single-agent inference-time frameworks can be highly non-trivial, with even leading strategies such as consensus requiring large increases in token costs. Overall, these results and observations appear to validate that the Conductor offers an impactful way to access higher performance not available to expensive existing single-agent frameworks, which is possible specifically through its powerful coordination strategies that compose the varying skills of different models.
>
> Table A1
>
> | Model                          | Performance | Token Usage | Cost      | Cost-adjusted performance |
> |--------------------------------|-------------|-------------|-----------|---------------------------|
> | Claude Sonnet 4 5x consensus   | 91          | 1412.8      | 0.021192  | 42.94073235               |
> | Claude Sonnet 4 5x reflect     | 90.66       | 2516.992    | 0.02080128| 43.58385638               |
> | Gemini 2.5 Pro 5x consensus    | 91.6        | 1658.4      | 0.016584  | 55.23396044               |
> | Gemini 2.5 Pro 5x reflect      | 88.33       | 2919.776    | 0.01675976| 52.70361867               |
> | GPT 5 5x consensus             | 91.3        | 1376.3      | 0.013763  | 66.33728112               |
> | GPT 5 5x reflect               | 91.79       | 2457.132    | 0.01424907| 64.41823923               |
> | Conductor                      | 93.14       | 735.2       | 0.009     | 103.4888889               |
>
>
>
> In addition, we also studied the performance-cost tradeoffs of the Conductor, not just compared with single-agent frameworks, but also with the other multi-agent frameworks. We present these additional performance-efficiency results in the following Table A2. Here, we recorded performance, token usage, and average cost of the Conductor alongside Smoothie, RouterDC, MoA, and MASRouter. Again, and in line with our findings in Figure 5 in the main text (the performance efficiency figure comparing average performance to API calls), we see the Conductor’s strong performance at relatively few API calls translates into substantial token and cost efficiency while delivering marked performance gains.
>
> Table A2
>
> | Model   | Performance | Token Usage | Cost |
> |----------------|---------|----------|------------------|
> | MoA            | 62.13 | 11203 |  $0.048554  |
> | Smoothie    | 56.48  |9909   | $0.039291  |
> | RDC            | 52.41  | 840    | $0.005613 |
> | MasRouter  | 56.89 | 4970   | $0.01345   |
> | Conductor   | 72.35 | 1820   | $0.02384   |
>
> As per the reviewer’s suggestion, **we have added all these additional performance-efficiency results to Appendix B of our revised manuscript**.
>
> [1] Wang, Xuezhi, et al. "Self-consistency improves chain of thought reasoning in language models." International Conference on Learning Representations (ICLR) (2023)

---

> > ### Comment · Reviewer_TwCE · 2025-11-27
> > **Response to the author**
> >
> > Thank you for the revisions and the detailed clarifications.
> > 1. The updated writing is much clearer. One remaining concern is the use of the term “latent capability.” Unlike concepts such as in-context learning or prompting, this term is not yet widely established in the literature, and the two cited papers do not explicitly use it. There is also a potential ambiguity with works on latent-space reasoning (e.g., "Coconut" Hao 2024, "V-JEPA" Assran 2024). It may be helpful to further clarify this terminology to avoid confusion.
> > 2. The extended baselines in Table B using stronger models are helpful. They clearly demonstrate that prompting a strong LM alone does not match the performance of a trained Conductor. The explanation regarding MoA and MASRouter is still somewhat unconvincing to me: benchmarks beyond LCB (such as the 30-question AIME set) also exhibit high variance, while these are baselines, an in-depth analysis is not strictly required within the scope of this paper.
> > 3. My three questions have been satisfactorily addressed.
> >
> > Overall, the rebuttal provides solid additional results and substantially improved clarity. I am convinced that this is a solid work, and I will raise my score to 6.

---

### Author Response · Authors · 2025-11-20
**Global Response**

Dear Reviewers and AC,

We thank the reviewers for taking the time to review our work and for their constructive feedback and suggestions. We’re particularly appreciative of the endorsements of our work, such as its strong empirical results including SOTA performance (all reviewers), the conceptual novelty of our framework (reviewers og9K, c7Bs, 5Z3X), and scope for interesting future directions through recursion and adaptivity (reviewers 5Z3X, c7Bs).

Two common themes emerged from reviewer responses, which we’d like to address in this global response. The first was the request for additional details regarding performance-efficiency analysis, in particular, adding token usage and average cost to our efficiency analysis. Following this request, we’ve added this information to our existing performance-efficiency comparison with multi-agent baselines. We also ran an additional single-agent inference-time framework, consensus [1], and added to our existing self-reflection inference-time baselines the performance-efficiency tradeoffs of GPT-5, Gemini 2.5 Pro, and Claude Sonnet 4 in comparison with our Conductor. These results are shown in Tables A1 and A2, where we find that the Conductor offers substantive performance and efficiency improvements over both single-agent inference-time scaling and multi-agent baselines.

The second was the request for additional ablations of the Conductor framework, where we 1) use a powerful frontier model as the Conductor to perform agent selection and subtask assignment, and 2) replace all agent workers with a single powerful frontier model. We present these results in Table B and C, where we find that the Conductor far surpasses these additional baselines, which we believe offers strong evidence of the Conductor’s learned ability to first discern the true underlying capabilities of the available agents, and then to intelligently combine and compose the differing agents to deliver performance beyond any individual worker across a wide range of tasks.

We provide the main analysis, results, and figures about these points and beyond in the individual reviewer responses for easier reference. In case there are any outstanding comments or suggestions following our rebuttal, we hope the reviewers will not hesitate to get back to us, and we will focus on addressing them right away.

Thanks,
Authors

Table A1

| Model                          | Performance | Token Usage | Cost      | Cost-adjusted performance |
|--------------------------------|-------------|-------------|-----------|---------------------------|
| Claude Sonnet 4 5x consensus   | 91          | 1412.8      | 0.021192  | 42.94073235               |
| Claude Sonnet 4 5x reflect     | 90.66       | 2516.992    | 0.02080128| 43.58385638               |
| Gemini 2.5 Pro 5x consensus    | 91.6        | 1658.4      | 0.016584  | 55.23396044               |
| Gemini 2.5 Pro 5x reflect      | 88.33       | 2919.776    | 0.01675976| 52.70361867               |
| GPT 5 5x consensus             | 91.3        | 1376.3      | 0.013763  | 66.33728112               |
| GPT 5 5x reflect               | 91.79       | 2457.132    | 0.01424907| 64.41823923               |
| Conductor                      | 93.14       | 735.2       | 0.009     | 103.4888889               |



Table A2

| Model   | Performance | Token Usage | Cost |
|----------------|---------|----------|------------------|
| MoA            | 62.13 | 11203 |  $0.048554  |
| Smoothie    | 56.48  |9909   | $0.039291  |
| RDC            | 52.41  | 840    | $0.005613 |
| MasRouter  | 56.89 | 4970   | $0.01345   |
| Conductor   | 72.35 | 1820   | $0.02384   |


Table B

| Model                     | LCB   | AIME | BigCodeBench | GPQA-D | Avg.    |
|---------------------------|-------|------|--------------|--------|---------|
| GPT-5 conduct 7 models   | 50.86 | 76.67 | 34.5       | 77.78 | 59.9525  |
| GPT-5 conduct  3 models           | 67.43 | 93.3 | 33.1         | 86.36  | 70.0475 |
| Gemini 2.5 Pro conduct 3 models   | 70.29 | 93.3 | 35.13        | 87.62  | 71.585  |
| Conductor                 | 83.93 | 93.3 | 37.86        | 87.5   | 75.6475 |

Table C

| Model                  | AIME | BigCodeBench | GPQA-D | Avg.        |
|------------------------|------|--------------|--------|-------------|
| Claude Sonnet 4        | 74.3 | 37.16        | 77.7   | 63.05333333 |
| Gemini 2.5 Pro         | 78.3 | 37.51        | 84.8   | 66.87       |
| GPT-5                  | 90.8 | 32.75        | 82.3   | 68.61666667 |
| Conductor with only GPT-5 | 93.33| 33.5         | 82.6   | 69.81       |
| Conductor              | 93.3 | 37.86        | 87.5   | 72.88666667 |


[1] Wang, Xuezhi, et al. "Self-consistency improves chain of thought reasoning in language models." International Conference on Learning Representations (ICLR) (2023)

---

### Meta-Review · Area_Chair_niX2 · 2026-01-06

**Summary:**

This paper introduces "The Conductor," a 7B model trained via reinforcement learning to manage and coordinate a team of larger, specialized models (like GPT-5 and Gemini) to solve difficult reasoning tasks. The reviewers were impressed by the paper's strong empirical results, which achieved state-of-the-art performance on benchmarks like AIME and LiveCodeBench, demonstrating that a small model can effectively "orchestrate" much larger ones. Initial concerns focused on the lack of cost and efficiency analysis, the need for stronger baselines (such as using a frontier model like GPT-5 as the conductor), and whether the gains were due to orchestration or just better prompting. Because the authors provided thorough evidence showing the Conductor is more efficient than standard methods and outperforms frontier-model planners, I recommend the paper be accepted.

**Reviewer Concerns:**

The authors successfully addressed the majority of the reviewers' technical concerns during the rebuttal. Specifically, they added a comprehensive performance-efficiency analysis that proved the Conductor uses fewer tokens and costs less than single-agent "consensus" strategies. They also included a "GPT-5 as Conductor" baseline, which showed that a specifically trained smaller model actually makes better coordination decisions than a raw frontier model. While some concerns remain regarding whether certain math and coding tasks are the best fit for a "multi-agent" approach, and some terminology like "latent capability" remains slightly vague, the authors' new qualitative examples of "planners" and "executors" working together effectively demonstrated the value of their framework.

**Reviewer Scores:**

The reviewers responded very positively to the rebuttal materials. Reviewer TwCE was the most convinced, raising their score significantly from a 2 (Reject) to a 6 (Weak Accept) after their questions on baselines and efficiency were answered. Reviewers og9K and 5Z3X maintained their high scores of 8, remaining confident in the work’s contribution to agentic workflows. Reviewer c7Bs remained the most skeptical with a 4, but given that the authors provided the specific "GPT-5 only" ablation they requested—which proved orchestration is better than just prompting—it is highly likely this reviewer would have moved toward a 5 or 6 had they participated in the final discussion.

---

### Decision · Program_Chairs · 2026-01-26

Accept (Poster)